# Photothermal catalytic transfer hydrogenolysis of protolignin

Hongji Li [1] ✉, Xiaotong Sun[1], Ting Li[2], Zhitong Zhao[3], Hui Wang[1], Xiaomei Yang[1], Chaofeng Zhang [2] ✉ & Feng Wang [4,5] ✉

Photothermal catalysis is a promising strategy to combine the advantages of both thermal-catalysis and photocatalysis. Herein we achieve the protolignin conversion to aromatics via the photothermal catalytic transfer hydrogenolysis process intensified by the in-situ protection strategy. The Pd/TiO$_2$ at 140 °C with UV irradiation can catalyze the reforming of primary alcohols to aldehydes and active H* species, which further participate in the acetalation protection of the 1,3-diol group of β-O-4 linkage and mediate the hydrogenolysis of C$_\beta$–OAr bonds, respectively. The conversion of birch sawdust with ethanol as the hydrogen donor provides a 40 wt% yield of phenolic monomers, compared with an 11 wt% monomer yield obtained from the conversion of extracted 1,3-diol-protected lignin under the same conditions. The synergistic effect of photocatalysis and thermal-catalysis contributes to the prior cleavage of the C$_\beta$–OAr bond before other C–O bonds. The feasibility of solar-light-driven photothermal catalytic system is demonstrated.

Lignin, the third largest biomass resource containing abundant functionalized aromatic structures, has received much attention as it can be potentially used to prepare the functionalized aromatic chemicals[1–4], which could reduce reliance on fossil-based resources. Focusing on the catalytic lignin conversion to chemicals, the key issue lies in the development of strategies and catalysts to selectively cleave the stable and ubiquitous C–O/C–C bonds, and meanwhile to keep the aromaticity unconverted[5–9]. In nature, the lignin, cellulose, and hemicellulose are intricately cross-linked to constitute the lignocellulose in wood and grass. Although the transformation of isolated lignin can avoid the interference of polysaccharides, the lignin extraction under harsh conditions can simultaneously cause the breakup of the intrinsic β-O-4 linkage and the nonnegligible condensation reactions via the formation of recalcitrant C–C bonds[10–12], which will prevent the further sufficient depolymerization of lignin. Therefore, compared with the isolated lignin conversion, the direct transformation of protolignin from raw lignocellulosic biomass to chemicals is appealing but more challenging, especially for the

preferential conversion of lignin and retention of cellulose based on the concept of lignin-first biorefinery[13–17].

Compared with oxidation[18,19], reduction[20,21], hydrolysis[22,23], and other depolymerization methods[24] or combined strategies[25–29], catalytic hydrogenolysis[30–38] including transfer hydrogenolysis[39–43] featured a higher yield of phenolic monomers. For example, Li[44] reported that the Ni-W$_2$C-mediated hydrogenolysis of birch powders at 235 °C and 6 MPa H$_2$ could efficiently deliver aliphatic diols (75.6 wt%) and monophenols (45.6 wt%). The high-temperature hydrogenolysis process has obvious advantages in converting all main components. Still, the goal of the lignin-first biorefinery with lignin preferential conversion requires further development of systems to regulate the hydrogenolysis capacity and selectivity. Compared with the high-temperature hydrogenolysis with excess H$_2$ to directly generate active H* on the catalyst surface, the H* generation through the hydrogen transfer mechanism involving a higher $E_a$ could be a choice[39–41]. We once reported that Ni/AC could catalyze fragmentation-hydrogenolysis of birch sawdust to provide a 50 wt% lignin conversion

[1]College of Chemistry, Zhengzhou University, 100 Science Avenue, Zhengzhou, China. [2]Jiangsu Co-Innovation Center of Efficient Processing and Utilization of Forest Resources, College of Light Industry and Food Engineering, Nanjing Forestry University, 159 LongPan Road, Nanjing, China. [3]College of Chemical Engineering and Technology, Taiyuan University of Technology, Taiyuan, China. [4]State Key Laboratory of Catalysis, Dalian National Laboratory for Clean Energy, Dalian Institute of Chemical Physics, Chinese Academy of Sciences, 457 Zhongshan Road, Dalian, China. [5]University of Chinese Academy of Sciences, Beijing, China. ✉e-mail: hongjili@zzu.edu.cn; zhangchaofeng@njfu.edu.cn; wangfeng@dicp.ac.cn

and a 97% selectivity toward phenolic monomers at 200 °C with methanol solvent under argon atmosphere[14], during which the typical lignin oligomers were firstly generated from the wood solvolysis and underwent further transfer hydrogenolysis with C−O bond cleavage to form phenolic monomers. The quick lignin release from solid biomass at 200 °C led to a high concentration of the lignin fragments[14], which was not only critical for further lignin depolymerization but also conducive to the corresponding condensation reactions among lignin fragments[45] (Fig. 1a). In addition, the high temperature could also cause high energy consumption and over-hydrogenation with aromatic ring breakdown. Although decreasing the temperature to 120-140 °C can ensure the lignin extraction from solid biomass, it will affect the generation and performance of the active H* on the catalyst surface, making the relatively low temperature unfavorable to the overall lignin transformation. One potential method to restrain the side reactions at a high temperature is employing the flowthrough system with the tubular reactor[46,47], which involves pre-extraction-fragmentation of lignin from solid biomass to lignin oligomers with hot alcohol and allows the reaction time of the following hydrogenolysis to be precisely controlled. However, to obtain a high reaction selectivity by controlling the reaction time before the total conversion, it is usually necessary to sacrifice a certain amount of conversion. As another method to regulate the reduction capacity, the semiconductor-mediated lignin photocatalytic hydrogen transfer depolymerization could be an alternative[48–54], during which the photo-generated holes ($h^+$) and electrons ($e^-$) can oxidize the hydrogen donor and reduce proton to the active H*, respectively. Nevertheless, the current photocatalytic transformation methods at the near-room temperature delivered a modest yield of monomers due to the difficult solvolysis of lignin or the bad collision contact between catalyst and protolignin (Fig. 1b), especially for the internal protolignin below the solid biomass surface, which requires a certain temperature to achieve the dissolution and initial fragmentation of lignin from the solid lignocellulose.

Given the above contradiction, one potential and simple improvement strategy is combining the advantages of photocatalytic systems and thermal catalytic systems, achieving the efficient extraction-fragmentation-conversion of protolignin by a photothermal catalytic transfer hydrogenolysis system at a relatively mild temperature with light irradiation. In the proposed photo-thermal system, a mild but high enough temperature can ensure the release of lignin, and light irradiation over the semiconductor catalyst can induce the oxidation of sacrifice reagent by the hole ($h^+$) and provide active H* species from proton and electron ($e^-$) for the consequent hydrogenolysis of lignin linkages, which can overcome the challenge in efficient generation of active H* under a mild condition for the thermal catalysis and the challenge in efficient extraction of lignin from solid wood powder during the traditional photocatalysis. At the same time, the release of lignin from lignocellulosic solids can be controlled to a certain extent by lowering the temperature appropriately, which may decrease the condensation of lignin fragments.

Here we show the photothermal catalytic transfer hydrogenolysis system for the protolignin conversion (Fig. 1c). After the reaction condition optimization, it is found that the Pd/TiO$_2$ could efficiently catalyze the transfer hydrogenolysis of protolignin in birch sawdust under a relatively mild condition with 370 nm light irradiation at 140 °C, which provides a 40 wt% monophenols (including 30 wt% yield of propyl-substituted phenols) by using primary alcohols as hydrogen donor. Compared to reported works on lignin and protolignin valorization via transfer hydrogenolysis, this photothermal system delivers a close to theoretical yield of monomers under milder conditions (Supplementary Table S1, for more information). At the same time, the roles of the primary alcohol are proposed, which include the solvent in combination with dioxane for the lignin extraction, the hydrogen donor for the photothermal

catalytic transfer hydrogenolysis of released lignin, and the precursor of the aldehyde as the protecting agent of diol structures in lignin. Besides, the released lignin from the mild in-situ solvolysis has a lower molecular weight than externally extracted lignin, which makes the conversion of the in-situ extracted protolignin more efficient than the extracted lignin under the same conditions. Furthermore, the synergistic effect of photocatalysis and thermal catalysis in this process is supported by controlled experiments. The heating not only mediates the extraction-fragmentation of lignin but also affects the photocatalytic reaction mechanism of β-O-4 linkage cleavage. In brief, we realize the efficient photothermal catalytic transfer hydrogenolysis of protolignin to aromatics intensified by the in-situ extraction-protection strategy, which could overcome the shortcomings of single thermal catalysis or photocatalysis.

## Results

### Photothermal catalytic transfer hydrogenolysis of lignin β-O-4 model

According to previous reports[10–12], the aldehyde additive can protect the 1,3-diol structure from condensation and greatly increase the monomers yield in the lignin hydrogenolysis. In our proposed photothermal system, the aldehyde can be generated from the photocatalytic oxidation of primary alcohol when the alcohol is utilized as the hydrogen donor, so the aldehyde-mediated 1,3-diol protection and selective lignin hydrogenolysis could be realized in one pot. Based on this conjecture, the photothermal catalytic transfer hydrogenolysis of the lignin β-O-4 diol model **1** was tested under 370 nm light irradiation at 140 °C, during which HCl was added to accelerate the potential diol acetalation mediated by the acetaldehyde from the hydrogen donor-ethanol (Fig. 2a). The acetal product **2** was then observed when TiO$_2$ was used as the photocatalyst (Fig. 2a, entry 1), which verified the feasibility of 1,3-diol protection under photothermal catalytic conditions. To further accelerate the generation of active H* species for the β-O-4 ether linkage cleavage, the metal cocatalysts including Co, Ni, Pt, and Pd particles were loaded on the TiO$_2$ surface, respectively. It was found that the Co, Ni, and Pt cocatalysts just promoted the formation of **2** (Fig. 2a, entries 2–4) and did not initiate the critical C$_\beta$−OAr bond cleavage, which could be attributed to their ability to promote the ethanol dehydrogenation to acetaldehyde and then the acetalation of diol **1** to acetal **2** was accelerated. Besides ethanol, the utilization of other primary alcohols (1-propanol and 1-butanol) as the hydrogen donor could also provide corresponding acetal products using Pt/TiO$_2$ as the photocatalyst (Supplementary Figs. S1 and S2). However, when the Pd particle (3 wt%) was used as the co-catalyst (Fig. 2a, entry 5), the transfer hydrogenolysis of **1** with C$_\beta$−OAr bond cleavage was aroused, which provided a 78% yield of n-propylbenzene **3** and a 67% yield of guaiacol **4** as the main products. At the same time, a 20% yield of byproduct **5** without C$_\beta$−OAr bond cleavage but with benzylic C$_\alpha$−O bond hydrogenolysis cleavage was obtained. No other byproducts can be identified in the gas chromatography data (Supplementary Figs. S3 and S4). Following reaction optimization results showed that the yield of guaiacol **4** could be increased to 83% and the generation of byproduct **5** could be avoided by adding twice as much photocatalyst (Fig. 2a, entry 6). The structure and morphology characterizations (Supplementary Figs. S5–S7 and Supplementary Table S2) of the optimal catalyst indicated that the metallic Pd nanoparticles with a mean size of 5.1 nm were evenly dispersed on the surface of TiO$_2$ particles. Although the loading of Pd particles could induce visible light absorption and promote the charge-carriers separation (Supplementary Fig. S8), no visible light-responsive catalytic activity was observed for Pd/TiO$_2$ (Supplementary Table S3). Meanwhile, the photocatalytic transfer hydrogenolysis occurred only when Pd particles were loaded on semiconductors (ZnO, Nb$_2$O$_5$, CeO$_2$, and C$_3$N$_4$) rather than inert carriers (SiO$_2$ and ZrO$_2$) under UV or visible light irradiation (Supplementary Table S4). Meanwhile, just increasing the Pd loading amount on the TiO$_2$ failed to increase the products of C$_\beta$−OAr

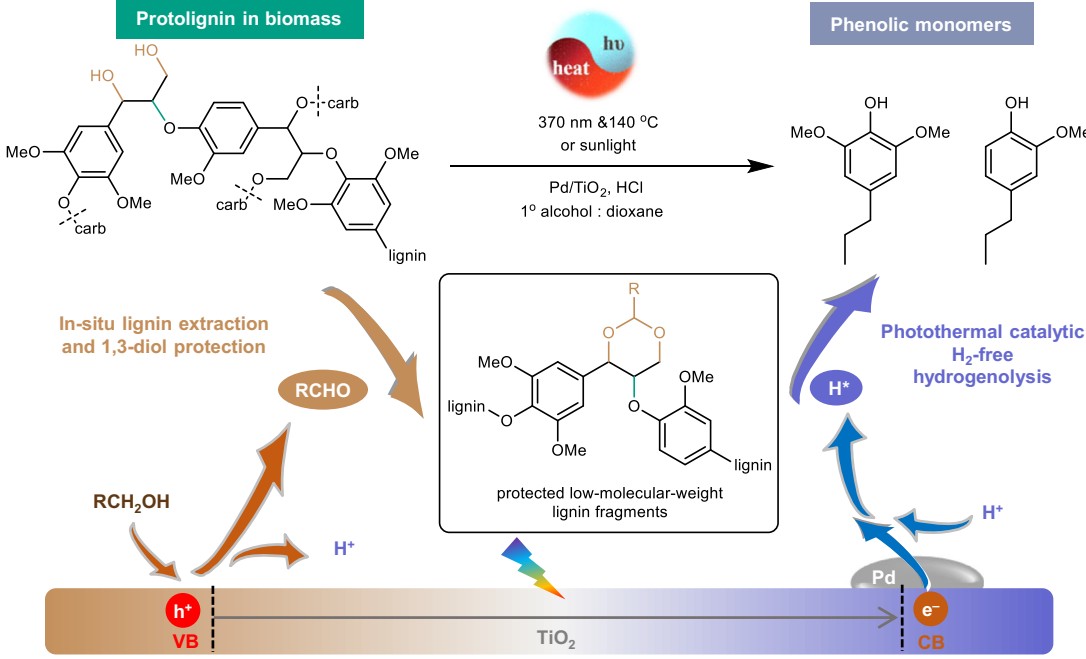

**Fig. 1 | A comparison of catalytic methods for the transfer hydrogenolysis of protolignin in lignocellulose. a** Thermal catalytic methods. **b** Photocatalytic methods. **c** Our developed photothermal catalytic method. The advantage and disadvantage are denoted as smile mark and sad marks, respectively.

bond cleavage (Fig. 2a, entry 7), which may show the critical synergistic effect between the semiconductor TiO$_2$ and loading Pd NPs. The higher Pd loading amount may not induce more exposed active sites due to the increase of Pd particle mean size to 8.1 nm (Supplementary Fig. S6). Meanwhile, the larger size of Pd particles could affect the light

absorption of semiconductor support and decrease the density of photo-generated electrons (Supplementary Fig. S8). Besides ethanol, other simple aliphatic alcohols also can be utilized as the hydrogen donor and solvent, delivering slightly lower yields of hydrogenolysis products (Supplementary Fig. S9).

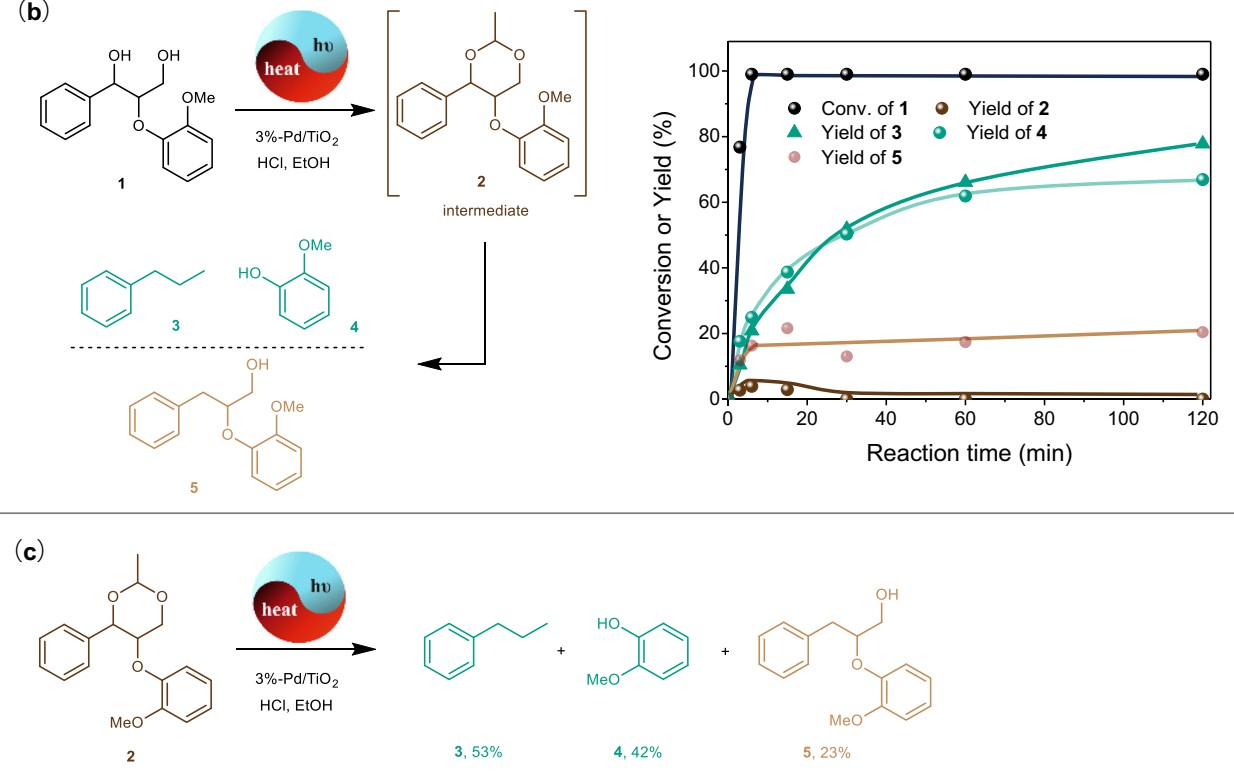

**(a)**

| Entry | Photocatalyst | Conversion (%) | Yield (%) | | | |
|---|---|---|---|---|---|---|
| | | | **2** | **3** | **4** | **5** |
| 1 | P25 TiO₂ | 20 | 4 | 0 | 0 | 0 |
| 2 | 3%-Co/TiO₂ | 66 | 43 | 0 | 0 | 0 |
| 3 | 3%-Ni/TiO₂ | 58 | 57 | 0 | 0 | 0 |
| 4 | 3%-Pt/TiO₂ | 72 | 50 | 0 | 0 | 0 |
| 5 | 3%-Pd/TiO₂ | 99 | 0 | 78 | 67 | 20 |
| 6ᵃ | 3%-Pd/TiO₂ | 99 | 0 | 78 | 83 | 0 |
| 7 | 6%-Pd/TiO₂ | 99 | 0 | 42 | 61 | 32 |

Condition: substrate (10 mg), photocatalyst (PC, 10 mg), EtOH (1 mL), HCl (37%, 10 µL), Kessil LED (370 nm), 140 °C, argon, 2 h. ᵃ PC (30 mg).

**Fig. 2 | The formation and transformation of 1,3-diol protected intermediate during the photothermal catalytic transfer hydrogenolysis of β-O-4 diol model compound. a** The photothermal catalytic transfer hydrogenolysis of β-O-4 1,3-diol model over different photocatalysts; **b** The time course of Pd/TiO₂ catalyzed transfer hydrogenolysis of model **1**. Reaction conditions: substrate **1** (10 mg), 3%- Pd/TiO₂ (10 mg), EtOH (1 mL), HCl (37%, 10 µL), Kessil LED (370 nm), 140 °C, Ar; **c** The conversion of 1,3-diol-protected intermediate **2** on Pd/TiO₂. Reaction conditions: substrate **2** (10 mg), 3%-Pd/TiO₂ (10 mg), EtOH (1 mL), HCl (37%, 10 µL), Kessil LED (370 nm), 140 °C, argon, 2 h.

In addition, according to the time course of Pd/TiO₂ catalyzed transfer hydrogenolysis of the β-O-4 diol model (Fig. 2b), it can be concluded that the acetal intermediate **2** was initially formed and rapidly transformed into hydrogenated products (**3, 4,** and **5**). The apparent quantum efficiency was 16.7% based on the proposed elementary reaction steps (Supplementary Fig. S10) and kinetic data obtained at the initial stage (Supplementary Fig. S11). Besides, when acetal **2** was utilized as the substrate (Fig. 2c), the product distribution was similar to that of **1**, which further verified that this photothermal

catalytic hydrogenolysis involved the in-situ protection of β-O-4 diol structure before its further reductive cleavage.

Meanwhile, the synergistic effect between photocatalysis and thermal catalysis played a critical role in this reaction (Fig. 3). Initially, this transformation diminished with decreasing of 370 nm light intensity and failed after switching the UV light to visible light (Supplementary Table S3), which suggests the UV-triggered photocatalysis was essential for the whole transformation. The Pd/TiO₂ system tended to catalyze the hydrogenolysis of $C_\alpha$–OH into $C_\alpha H_2$ (**5,**

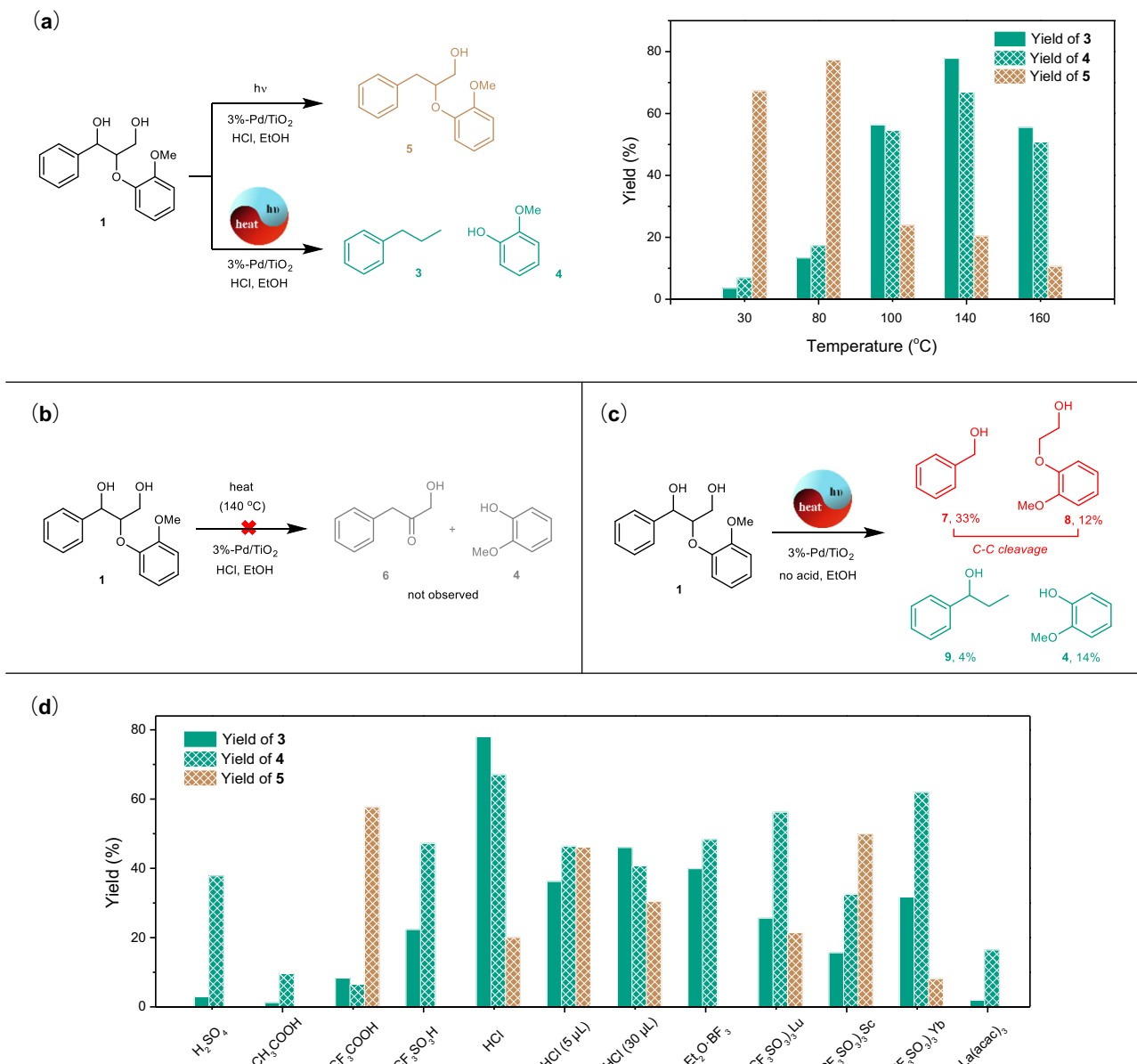

**Fig. 3 | The photothermal effects on the transfer hydrogenolysis of the lignin model. a** The effect of heating on the photocatalytic hydrogenolysis pathways. Reaction conditions: substrate **1** (10 mg), 3%-Pd/TiO₂ (10 mg), EtOH (1 mL), HCl (37%, 10 μL), Kessil LED (370 nm), 30–160 °C, argon, 2 h; **b** The controlled experiment under dark conditions; **c** The controlled photothermal catalytic reaction without acid; **d** The performance of different acids (liquid, 10 μL; solid, 10 mg) under the standard photothermal conditions.

67% yield) rather than the $C_\beta$–OAr ether bond cleavage at room temperature under UV light irradiation (Fig. 3a), which was consistent with our previous results that the benzylic $C_\alpha$–O bond was reactive on Pd/TiO₂ at room temperature[55,56]. However, the hydrogenolysis selectivity of the $C_\beta$–OAr bond rather than the conversion of the substrate without linkage cleavage was dramatically improved by increasing the reaction temperature under light irradiation (Fig. 3a). With the assistance of heat by raising the temperature to 140 °C, the Pd/TiO₂ system tended to catalyze the $C_\beta$–OAr ether bond cleavage and provided n-propylbenzene and guaiacol as the monomer products. Furthermore, considering that heat might not only promote the transfer hydrogenolysis but also trigger the acidolysis because of the HCl additive, the controlled experiment without light irradiation was performed at 140 °C, and the unobvious conversion result suggested that the pure thermal catalytic acidolysis and transfer hydrogenolysis of the lignin model did not occur in the photothermal conditions (Fig. 3b), which excluded the reaction

pathway of tandem thermal catalytic acidolysis followed by photocatalytic hydrogenation. For the promotion effect of the temperature on the photothermal system, further photocurrent characterization of the Pd/TiO₂ catalyst at 30, 94, and 140 °C indicated the photocurrent slightly increased with the temperature, which suggested that heating favored the separation of photo-generated charge carriers (Supplementary Fig. S12).

Further acid optimization with various Brönsted acids and Lewis acids showed the importance of the acid additive (Fig. 3d), especially for the HCl, which could efficiently promote the hydrogenolysis cleavage of the $C_\beta$–OAr linkage. In addition, when the acid additive was removed from the photothermal system (Fig. 3c), monomer products were generated in a much lower yield from the $C_\beta$–OAr cleavage, and $C_\alpha$–$C_\beta$ bond cleavage occurred simultaneously. Given the redox properties of the photothermal system, the benzyl alcohol may be generated from the hydrogenation of benzaldehyde, which could be derived from the β-scission of the $C_\alpha O\bullet$ fragment after the hydrogen

**(a) Controlled experiments**

**(b) Proposed reaction route of 1,3-diol model**

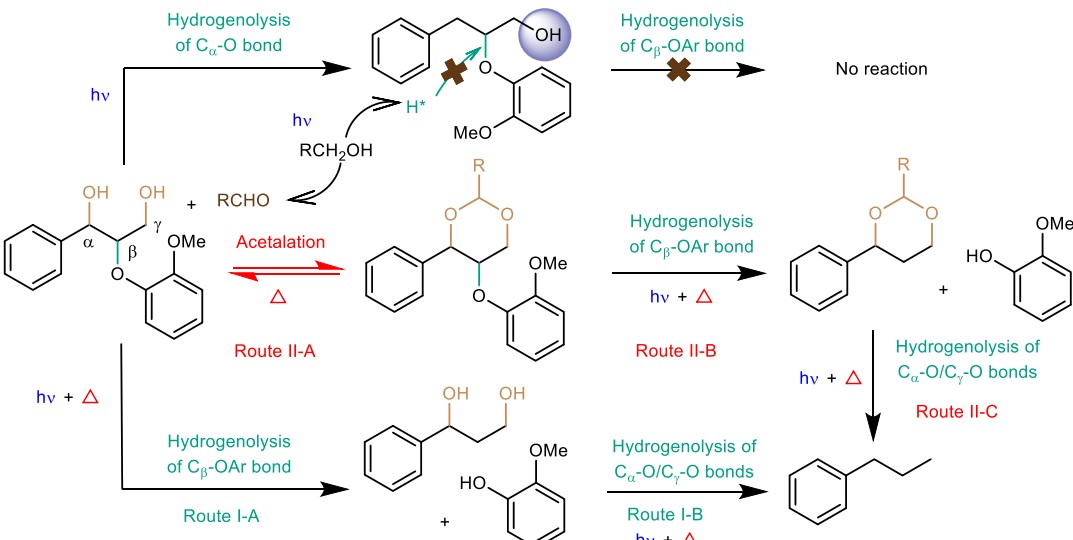

**Fig. 4 | Controlled experiments and proposed reaction routes of β-O-4 diol model. a** Controlled experiments. Standard conditions: substrate (10 mg), 3%-Pd/TiO$_2$ (10 mg), EtOH (1 mL), HCl (37%, 10 μL), Kessil LED (370 nm), 140 °C, argon, 2 h. **b** Proposed reaction routes.

atom transfer (HAT) reaction between the 1,3-diol **1** and photo-generated holes (h$^+$)[4].

To further study the photothermal process focusing on the relevant bonds cleavage and hydrogenolysis pathway of β-O-4 linkage, controlled experiments with different substrates were performed. Compared with the normal β-O-4 diol model **1**, the β-O-4 dimer model **5** without the benzylic hydroxyl group showed no conversion (Fig. 4a, i), while the simultaneous absence of benzylic hydroxyl and C$_\gamma$−OH in β-O-4 dimer model **10** could facilitate the hydrogenolysis cleavage of the C$_\beta$−OAr bond (Fig. 4a, ii), delivering close to equivalent cleaved products. Based on these results, it can be deduced that the presence of benzylic hydroxyl is not essential to the cleavage of the C$_\beta$−OAr bond, but the single C$_\gamma$−OH existence may inhibit the hydrogenolysis of the C$_\beta$−OAr bond in the β-O-4 dimer. In addition, acetal **2**

could be generated from the reaction between 1,3-diol **1** with acetaldehyde at room temperature and 140 °C, and the heating conditions were beneficial to the fast formation of acetal **2** (Fig. 4a, iii). In addition, when the potential C$_\beta$−OAr cleavage products including 4-phenyl-1,3-dioxane **13** and 1-phenylpropane-1,3-diol **14** were added into the photothermal catalytic system, both of them were efficiently converted into propylbenzene via hydrogenolysis of C$_\alpha$-O/C$_\gamma$−O bonds (Fig. 4a, iv). Therefore, the hydrogenolysis of β-O-4 models including the 1,3-diol model and protected model may undergo the direct cleavage of C$_\beta$−O bond followed by hydrogenolysis of C$_\alpha$−O/C$_\gamma$−O bonds (Fig. 4b). Based on the obtained results, there could exist two reasonable routes for the photothermal catalytic transfer hydrogenolysis of the β-O-4 diol dimer. The first route involves the direct reductive cleavage of the C$_\beta$−OAr bond and following hydrogenolysis of other

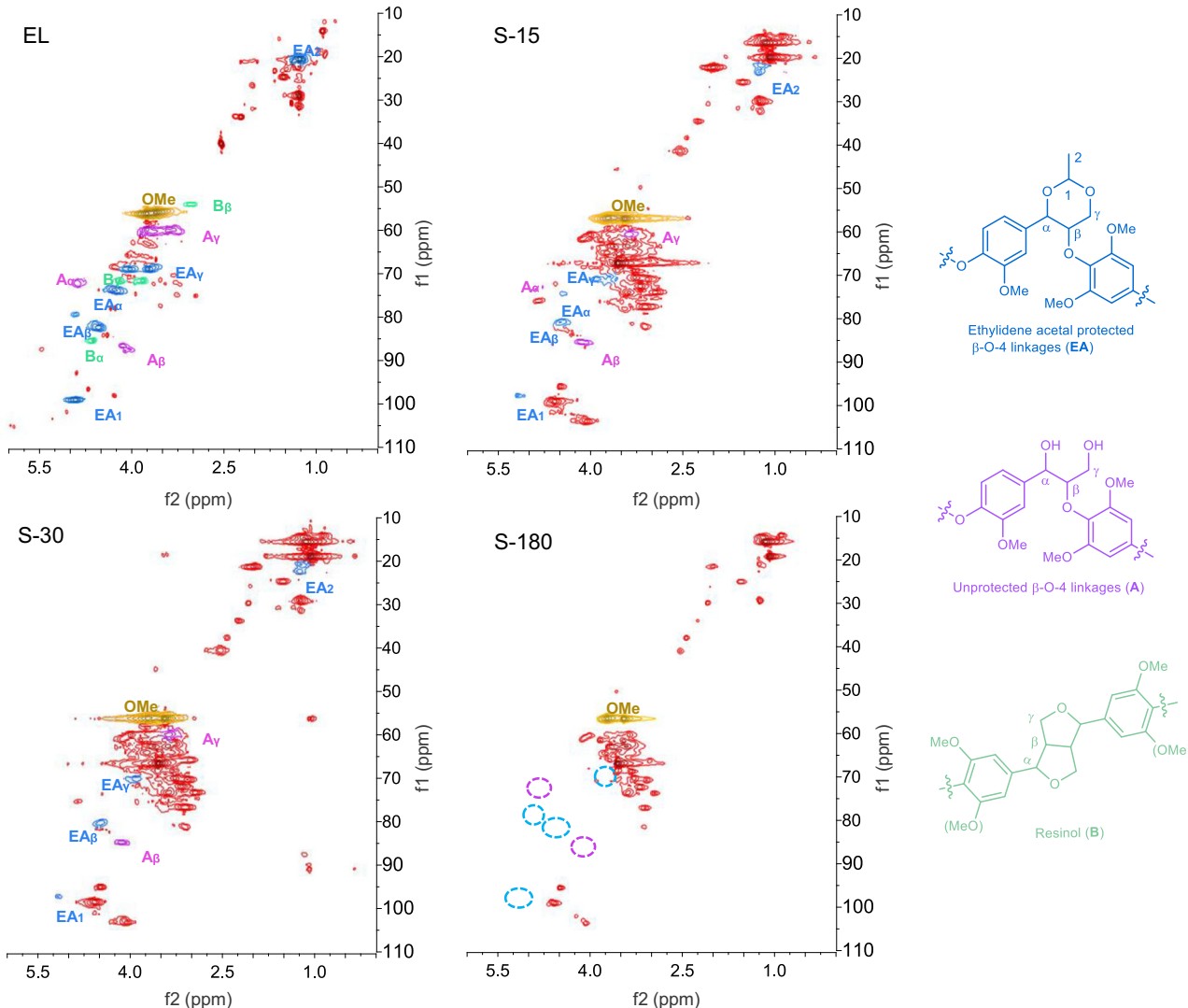

**Fig. 5 | The 2D HSQC NMR spectra of external extracted birch lignin with 1,3-diol protection (EL) and soluble lignin samples at 15, 30, and 180 minutes during photothermal catalytic transfer hydrogenolysis of birch sawdust (S-15,** **S-30 and S-180).** Reaction conditions of lignin transformation: birch sawdust (60 mg), 3%-Pd/TiO$_2$ (10 mg), EtOH (1 mL), dioxane (0.6 mL), HCl (37%, 10 µL), Kessil LED (370 nm), 140 °C, argon.

C−O bonds under the photothermal synergistic catalysis with ethanol as the hydrogen donor (Fig. 4b, route I). The second route was the tandem process consisting of photocatalytic oxidation of primary alcohols, acetalation of the 1,3-diol structure of β-O-4 motif with aldehyde, photothermal hydrogenolysis cleavage of C$_β$−OAr bond, and further hydrogenolysis of monomer products (Fig. 4b, route II).

**Photothermal catalytic transfer hydrogenolysis of birch protolignin**

Then the photothermal catalytic transfer hydrogenolysis method was applied for the conversion of the solid birch sawdust, which was composed of 19.2 wt% lignin, 38.6 wt% cellulose, and 23.8 wt% hemicellulose as determined by the NREL method. Different from the β-O-4 model conversion, the addition of 1,4-dioxane besides the ethanol as solvent was proposed to extract lignin in situ during transfer hydrogenolysis. Firstly, the 2D HSQC NMR spectra of in-situ extracted lignin samples during hydrogenolysis were recorded and compared with the externally extracted 1,3-diol-protected lignin sample which was prepared according to previous works (Fig. 5)[10,11]. The ethylene acetal-protected β-O-4 linkage (blue regions) was identified in the NMR spectra of externally extracted lignin (Fig. 5, EL), in combination with a small amount of unprotected β-O-4 linkage (purple regions) and

resinol linkage (green regions). The ethylidene acetal protected β-O-4 linkage was also found in the in-situ extracted lignin samples at 15 and 30 min (Fig. 5, S-15 and S-30), proving the 1,3-diol protection of lignin fragments under photothermal catalytic conditions. Furthermore, both protected and unprotected β-O-4 linkages diminished after 3 hours of reaction (Fig. 5, S-180), suggesting the hydrogenolysis of these linkages.

Then, the effect of in-situ extraction and protection on lignin transfer hydrogenolysis was embodied in the yield of monomers using different alcohols as the hydrogen donor (Fig. 6a, Supplementary Figs. S13 and S14). The monomer yields obtained in primary alcohols (30-42%) were much higher than those obtained in secondary alcohols (16-18 wt%). For example, the yields of 2,6-dimethoxy-4-propylphenol and 2-methoxy-4-propylphenol in 1-propanol/1,4-dioxane were 23 wt% and 7 wt%, respectively. However, when 1-propanol was replaced by 2-propanol, the conversion just provided a 7 wt% yield of 2,6-dimethoxy-4-propylphenol and a 1 wt% yield of 2-methoxy-4-propylphenol. This phenomenon was contrary to the hydrogenolysis result of the lignin model **1** in different alcohols (Supplementary Fig. S9). Even the photocatalytic dehydrogenation product ketone from secondary alcohol was a modest protection reagent, the secondary alcohol was a better hydrogen donor than

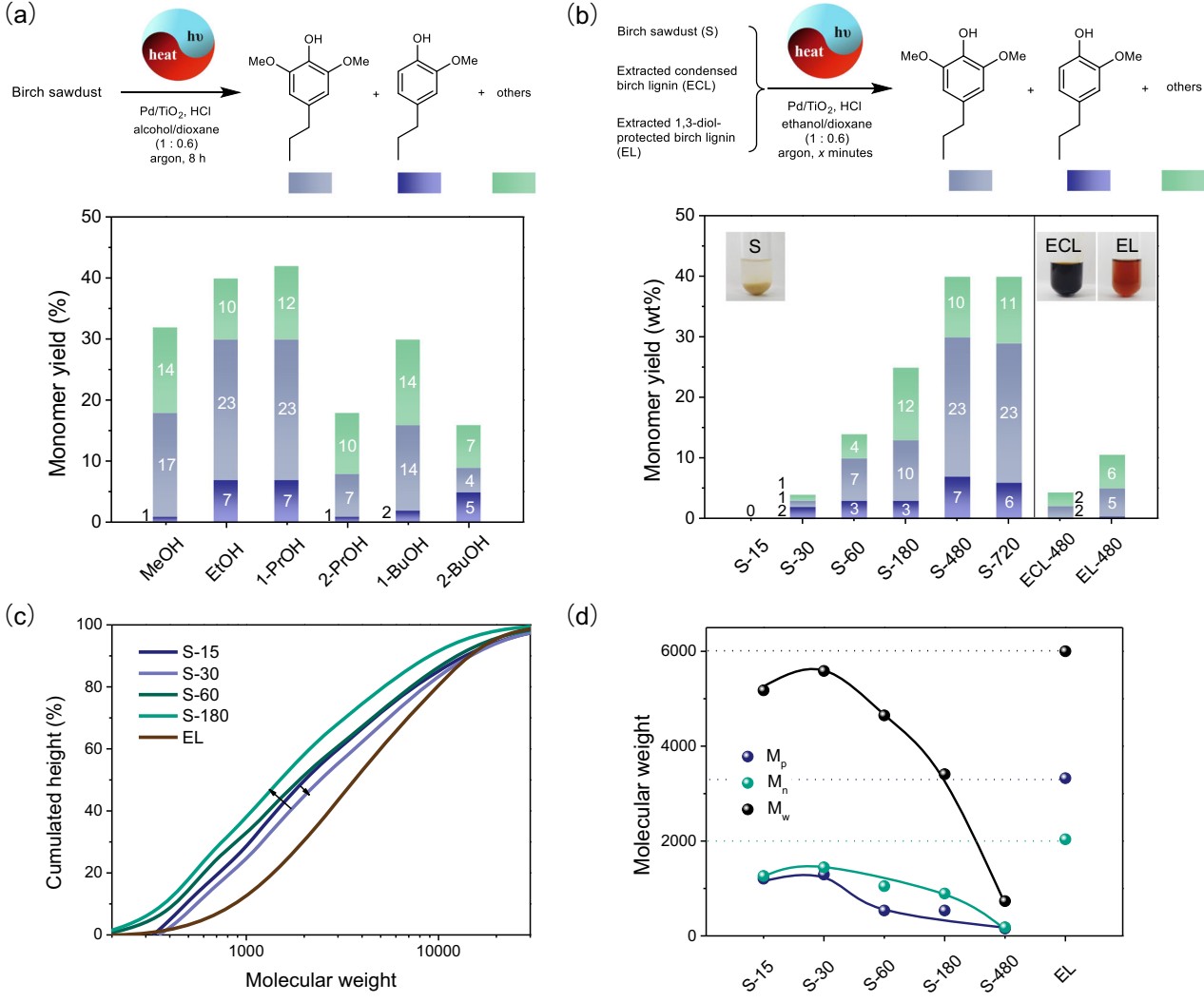

**Fig. 6 | The photothermal catalytic transfer hydrogenolysis of lignin samples.**
**a** The transformation of birch sawdust into phenolic monomers using different alcohols as the hydrogen donor. Birch sawdust (60 mg), 3%-Pd/TiO$_2$ (10 mg), alcohol (1 mL), dioxane (0.6 mL), HCl (37%, 10 µL), Kessil LED (370 nm), 140 °C, argon, 8 h; **b** The transformation of birch sawdust at different reaction times (15, 30, 60, 180, 480, and 720 min, the samples are denoted as S-15, S-30, S-60, S-180, S-480, and S-720) and the transformation of extracted condensed lignin (ECL) and 1,3-diol-protected lignin (EL) samples for 480 min in ethanol/dioxane. The illustrations are initial wood/lignin samples dispersed in ethanol/dioxane (v/v, 1:0.6); **c**, **d** The molecular weight analysis of extracted 1,3-diol-protected lignin and soluble lignin samples during the transformation of birch sawdust by GPC. M$_p$, molecular weight of the peak maxima; M$_n$, number average molecular weight; M$_w$, weight average molecular weight. Conditions for **b**, **c**, and **d**: birch sawdust (60 mg) or lignin samples (10 mg), 3%-Pd/TiO$_2$ (10 mg), ethanol (1 mL), dioxane (0.6 mL), HCl (37%, 10 µL), Kessil LED (370 nm), 140 °C, argon.

primary alcohol. Besides, the hydrogenolysis performance of the ethylidene acetal-protected model was slightly worse than that of the unprotected model (Fig. 2b, c). Thus, the ability of hydrogen donating rather than protecting effect dominated the model conversion. In contrast, the higher yields of monomers from protolignin conversion in primary alcohol with modest hydrogen donating ability demonstrated the importance of protecting effect in this tandem extraction/hydrogenation process of real biomass[11]. These results also indicated the difference between lignin model transformation with protolignin transformation.

Furthermore, although the in-situ extraction and 1,3-diol protection can efficiently promote the depolymerization of protolignin in birch sawdust, the direct transfer hydrogenolysis of externally extracted birch lignin with or without 1,3-diol protection just delivered monomers in yields of 11 wt% and 4 wt% in the same photothermal catalytic system which was much lower than that from birch wood conversion with a 40 wt% yield of phenolic monomers at the same

reaction time of 480 min (Fig. 6b). Then, the GPC characterization focusing on the molecule weight change was carried out to reveal the reason. The results indicated the molecular weight of in-situ extracted lignin increased before 30 minutes and then decreased gradually (Fig. 6c, d), which suggested that the timely conversion of in-situ extracted lignin could restrain the recondensation of fragments. At the same time, all the in-situ extracted lignin samples showed a lower molecular weight distribution than the externally pre-extracted lignin with 1,3-diol protection (EL) (Fig. 6c, d). Given the fact that the releasing of protolignin from wood was a slow and gradual process and the released lignin macromolecules could be depolymerized to the smaller ones efficiently, the lignin concentration during the in-situ extraction-protection-hydrogenolysis should be lower than the solution of the pre-isolated lignin. Therefore, the better performance of in-situ extracted lignin than the pre-isolated lignin is not only because of the decrease of lignin fragment size favoring its transformation over the heterogeneous catalyst but also related to the controllable lignin

release and low lignin concentration that can reduce the occurrence of undesired fragments condensation.

In addition, HCl may play multiple roles during the photothermal catalytic conversion of protolignin (Supplementary Fig. S15). The above tests on the lignin model have proved the promoting effect of HCl on the selective ether bond cleavage (Fig. 3C). As for the lignocellulosic substrate, the lignin was barely extracted from birch sawdust without HCl under the heating conditions. Therefore, HCl was vital for the in-situ lignin extraction. Besides, the condensation of lignin 1,3-diol motif with acetaldehyde can be improved by HCl. However, HCl has a negative effect on the hemicellulose and even cellulose retaining (*vide infra*). The photothermal experiment of birch sawdust without the addition of photocatalysts but with HCl indicated that hemicellulose was totally decomposed and cellulose was partly decomposed (Supplementary Table S6).

### The recycling of solid photocatalysts

Given the fact that the usage of a solid heterogeneous catalyst in solid biomass transformation usually meets the challenges in the separation of catalyst from the post-reaction mixture and the deactivation of the active catalyst, the recycling of solid photocatalyst in this photothermal system was further checked. After the photothermal reaction, both lignin and hemicellulose were depolymerized into small molecules that were soluble in the liquid phase (the analysis of products from hemicellulose, see Supplementary Table S6), leaving the cellulose mixed with the photocatalyst in the solid phase (Fig. 7a). The first solvent removal in the liquid phase and the following mixture redissolution in water could separate hemicellulose products from lignin products. The isolation of the photocatalyst was performed via acid-catalyzed hydrolysis of cellulose, which delivered a 59 wt% yield of glucose in the aqueous solution. The solid residue underwent further calcination in air to achieve photocatalyst recycling. The comparison of TEM images (Fig. 7b, c, e, f) and XPS spectra (Fig. 7d, g) indicated that the loaded Pd particles aggregated and transformed into $PdO_x$ species after the calcination regeneration. Nevertheless, the photocatalyst could be used for another four cycles (Fig. 7h), during which the $PdO_x$ could be in-situ reduced in the transfer hydrogenolysis system. Therefore, the cycle stability of $Pd/TiO_2$ can decrease the cost of precious metal in further large-scale biomass utilization.

### Catalytic conversion of protolignin in other lignocellulosic biomass

To explore the application scope of this $Pd/TiO_2$-catalyzed photothermal system in the protolignin depolymerization, other inedible lignocellulosic biomass including pine sawdust, bark of Platanus orientalis Linn, walnut shell, reeds straw, wheat straw, and corn straw powders were also tested in the photothermal catalytic system (Table 1). Before the depolymerization test, the lignin content of the selected biomass samples was determined according to the NREL method (Supplementary Table S7), which showed that the lignin contents in the selected wood biomass and reed straw were around 20 wt% and the corn and wheat straws have a lower lignin content of 13 wt% and 7 wt%, respectively. During the following photothermal catalytic transformation, the yields of phenolic monomers from the selected biomass samples were 17–42 wt% based on the weight of initial lignin in the biomass substrate (Table 1), and the main products from various biomass substrates were propyl-substituted phenols, which further verified the developed photothermal catalytic system being a promising method to upgrade the protolignin in diverse biomass to valuable aromatic chemicals.

**Photothermal transfer hydrogenolysis of proto-lignin driven by pure solar light**. Although the current photothermal system can efficiently promote protolignin depolymerization with a close to ideal yield of phenolic monomers, the above transformation system still needs extra electrical energy input to drive the heating device, LED light, and stirrer device (Fig. 8). Given the fact that solar energy is one typical renewable primary energy that can synchronously provide the necessary heat and irradiation conditions, we then used a Fresnel lens ($300 \times 300\,mm^2$) to concentrate solar light and provide sufficient temperature and irradiation for the transfer hydrogenolysis of birch protolignin (Fig. 8). By simply adjusting the distance and angle between the lens and the reaction tube to maintain the reaction at 140 °C for 6 h, the outdoor test of birch based on the pure solar energy provided a 34 wt% yield of phenolic monomers from birch sawdusts, including a 16 wt% yield of 2,6-dimethoxy-4-propylphenol and a 9 wt% yield of 2-methoxy-4-propylphenol. Furthermore, the performance of the concise pure solar-light-driven system in the transformation of other lignocellulosic biomass, including walnut shell, reed straw, wheat straw, and corn straw, was also slightly inferior to those of electrical-driven devices, but the obtained results demonstrated the feasibility of utilizing greener solar energy to achieve the photothermal catalytic transformation of protolignin. The utilization of solar light makes our method environmentally friendly and economically viable, which could provide guidance for further large-scale and low-cost lignin transformation.

### Techno-economic analysis and life cycle assessment (LCA)

An industrial-scale process flow has been designed and simulated to evaluate the practical potential of 4-propylguaiacol production from the solar-driven photothermal conversion of wheat straw, which featured the extensive source feedstock and high selectivity of a single product. The model employed wheat straw as feedstock with 100,000 metric tons per annual (t/a), which approximates 2800 t/a of 4-propylguaiacol yield. With feedstock pretreatment, photothermal reaction, consumable recovery, 4-propylguaiacol purification, cellulose hydrolysis, and photocatalyst recovery units (Supplementary Fig. S16), the model was performed through steady-state simulation using Aspen plus to obtain material and energy inventories according to current reaction conditions and parameters (Supplementary Tables S8 and 9). Techno-economic analysis and life cycle carbon emissions were subsequently carried out to identify the opportunity for industrialization. The results reveal that the total production cost (TPC) of 4-propylguaiacol can be lower at 44354 CNY/t, and consumables and depreciation costs are two constraints due to the small scale designed based on the biomass supply and product demand (Supplementary Table S10, Supplementary Fig. S17). Life cycle greenhouse gases (GHG) emission is 8.70 t $CO_2eq/t$ 4-propylguaiacol, of which wheat straw accounts for the largest contribution of 85.28% (Supplementary Figs. S18 and 19). The life cycle GHG emissions of wheat straw approximates 170 kg $CO_2eq/t$ due to plant, harvest, and transport. High wheat straw input causes more GHG emissions than the carbon credit of bio-derived chemicals, thus a higher conversion ratio is attractive to facilitate this work.

## Discussion

In conclusion, focusing on the efficient protolignin conversion in the lignocellulosic biomass to aromatic monomers, we herein developed a photothermal catalytic method to achieve protolignin in-situ extraction, 1,3-diol acetalation protection, and transfer hydrogenolysis to phenolic monomers in one pot. After the catalyst screening and reaction condition optimization, it was found that the $Pd/TiO_2$ at 140 °C with 370 nm light irradiation can efficiently catalyze the reforming of primary alcohols to the corresponding aldehydes and active H* species, which can further participate in the acetalation protection of the 1,3-diol group in the in-situ extracted β-O-4 linkage and mediate the following selective hydrogenolysis of the $C_\beta$−OAr bonds to aromatic monomers, respectively. According to the controlled experiments, the synergistic effect of photocatalysis and thermal catalysis is crucial to the prior cleavage of the $C_\beta$−OAr bond, avoiding the formation of an inert

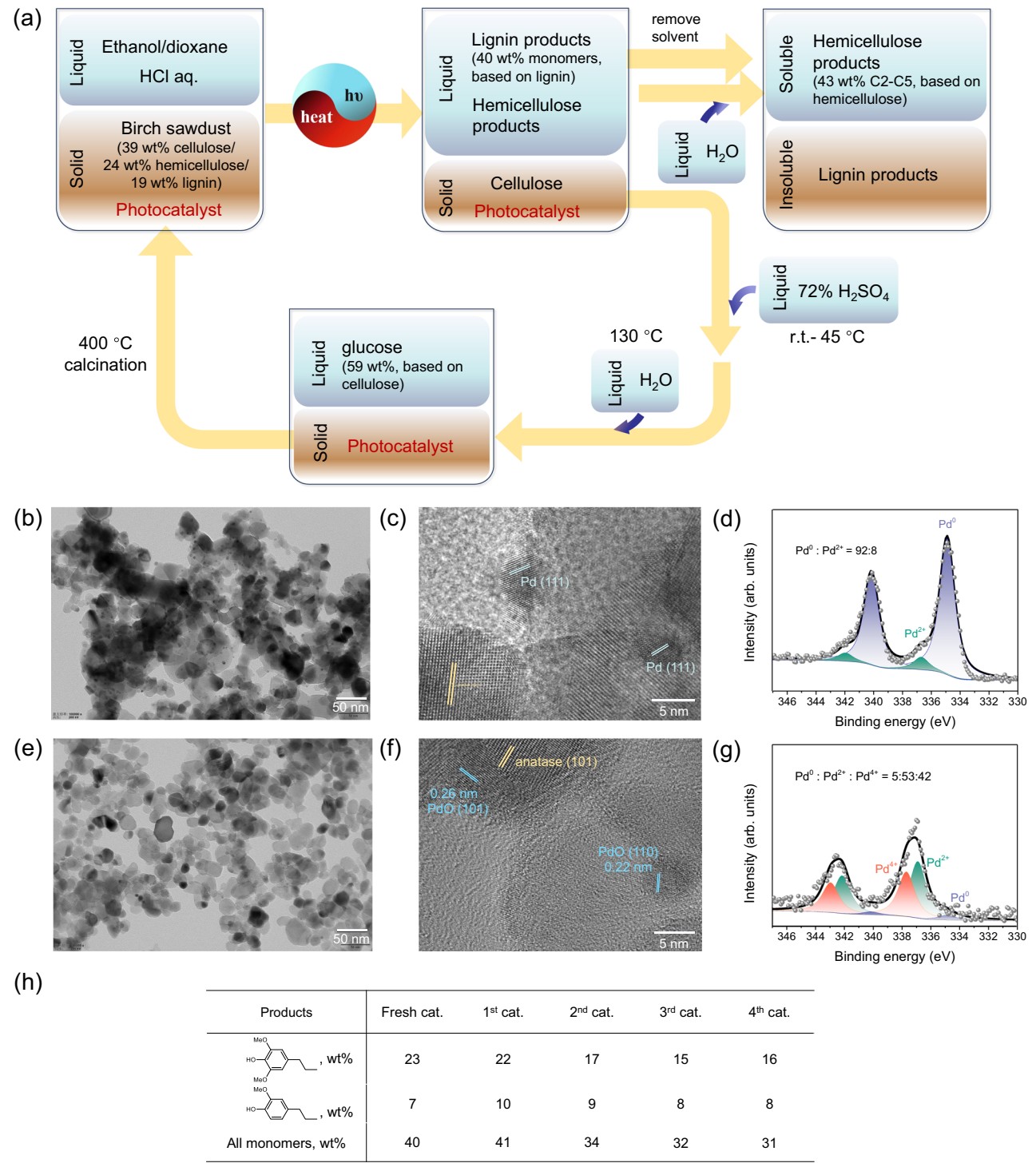

**Fig. 7 | The procedure of photocatalyst recycling and characterizations of Pd species in fresh and recycled photocatalysts.** The photocatalyst recycling procedure and transformations of lignin, hemicellulose, and cellulose (**a**); TEM images of fresh Pd/TiO$_2$ (**b**, **c**) and recycled Pd/TiO$_2$ (**e**, **f**); Pd *3d* XPS results of fresh Pd/TiO$_2$ (**d**) and recycled Pd/TiO$_2$ (**g**); and the catalytic performance of the recycled photocatalyst (**h**). The catalysts which undergo one, two, three and four cycles are denoted as 1st cat., 2nd cat., 3rd cat., and 4th cat.

benzylic dehydroxylated byproduct to prevent the C$_\beta$−OAr bond cleavage. Furthermore, the slow or controllable in-situ release of low-molecular-weight lignin fragments during biomass conversion benefited the lignin hydrogenolysis, which makes the whole biomass a better starting material than the externally isolated lignin. This method allowed the hydrogenolysis of lignin in various biomass into propyl-substituted monomers at 140 °C under H$_2$-free conditions. In addition, this work also applied the developed system in the conversion of lignin from more kinds of biomass resources and studied the recycling and reuse of the Pd/TiO$_2$ catalyst. The pure solar-light-driven photothermal catalytic transfer hydrogenolysis of protolignin was feasible without the extra electric energy input, showing its potentiality in large-scale applications. Although more efforts are still needed to update the simple photothermal system, the relevant photothermal research can further expand methods for converting and utilizing lignocellulosic biomass in the future.

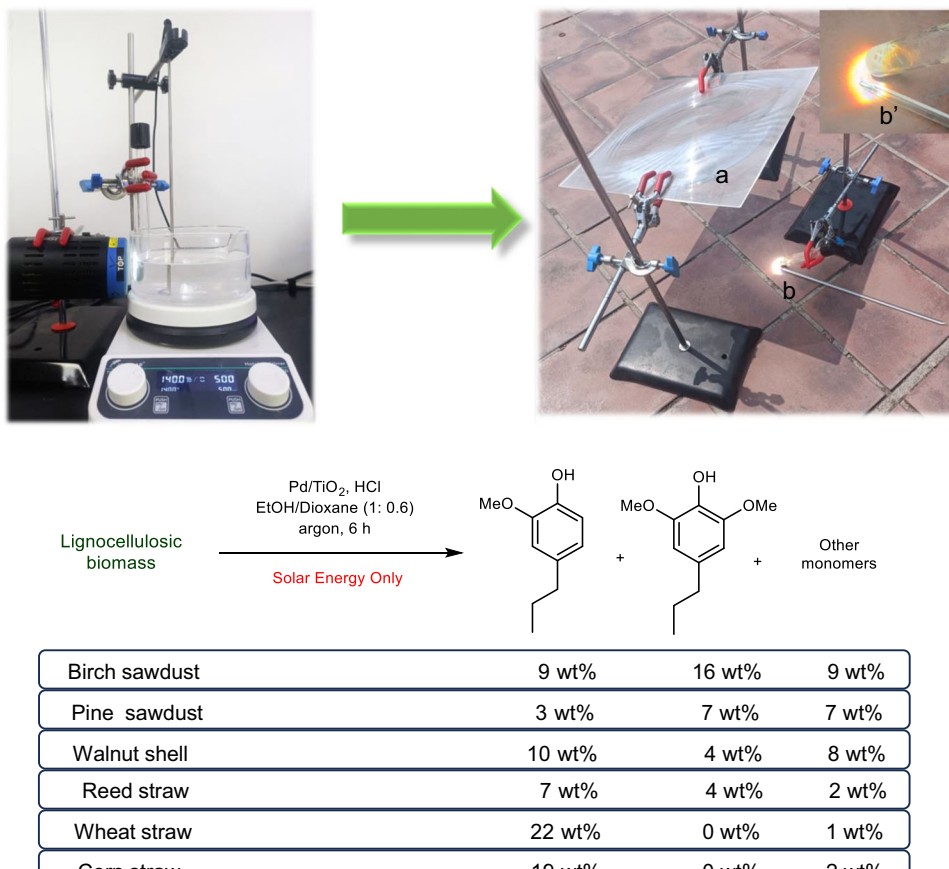

| Birch sawdust | 9 wt% | 16 wt% | 9 wt% |
|---|---|---|---|
| Pine sawdust | 3 wt% | 7 wt% | 7 wt% |
| Walnut shell | 10 wt% | 4 wt% | 8 wt% |
| Reed straw | 7 wt% | 4 wt% | 2 wt% |
| Wheat straw | 22 wt% | 0 wt% | 1 wt% |
| Corn straw | 19 wt% | 0 wt% | 2 wt% |

**Fig. 8 | The changing of driving energy for the photothermal transfer hydrogenolysis of biomass substrates from electric energy (left) to pure solar light (right), and the performance of pure solar energy system.** Reaction conditions: biomass (60 mg), Pd/TiO₂ (10 mg), HCl (37 wt%, 10 µL), EtOH/dioxane (5:3, 1.6 mL), argon, sunlight irradiation collected with a Fresnel lens. GC yields. Picture notes: (**a**) Fresnel lens 300 × 300 mm²; (**b**) Reaction tube with a thermometer; (**b'**) Boiling reaction solution at 140 °C maintained by controlling the light focusing.

## Methods

### The preparation of photocatalyst
All photocatalysts were prepared via the impregnation-reduction method. As for the 3%-Pd/TiO₂, TiO₂ (0.1 g, Degussa P25) was added into a 5 mL methanol solution of palladium (II) acetate containing 3 mg of Pd. After ultrasonic treatment for 30 min, the mixture was stirred for another two hours. The solvent was removed under vacuum, and the collected powder was calcined in H₂ flow (20 mL·min⁻¹) at 400 °C (heating rate 10 °C·min⁻¹) for 4 h. The preparation procedures of other metal/TiO₂ samples were similar to that of Pd/TiO₂ except for using Co(NO₃)₂·6H₂O, NiCl₂·6H₂O, and H₂PtCl₆·6H₂O as metal precursors.

### Characterizations of lignin samples and catalysts
Transmission electron microscopy (TEM) was carried out using a Jem-2100F electron microscope with an accelerating voltage of 200 kV. X-ray photoelectron spectroscopy (XPS) was collected on Thermo-fisher escalab 250xi. ICP-OES was performed on Thermo Fisher iCAP 7400. The ¹H-, ¹³C-, and 2D HSQC NMR spectra of compounds and lignin samples were measured on a Bruker AVIII 400 spectrometer (¹H: 400 MHz, ¹³C: 101 MHz). The Gel Permeation Chromatography (GPC) was measured on Agilent GPC 50.

### The procedure of the photothermal catalytic test
In a typical procedure, a 20 mL quartz reactor equipped with a stir bar was loaded with lignin model (10 mg) or wood sawdust (60 mg), photocatalyst (10 mg), acid (10 µL) and solvent (1.0 or 1.6 mL). Then the atmosphere was switched to argon before sealing the reactor. This mixture was irradiated using Kessil PR160L LED lamps (λ_max = 370 nm,

40 W) for a certain time. After the reaction, the standard solution (naphthalene in ethanol) was added. After filtration using a Nylon syringe filter, the solution was analyzed by gas chromatography (Shimadzu GC-2014C) and GC/MS (GC: Shimadzu 2030AM, MS: Shimadzu, QP2020NX).

The experimental procedure of protolignin transformation using solar energy was similar to the above method except for using the combination of sunlight with a Fresnel lens to replace the LED lamp and the heating device.

All aromatic products were quantified using a GC (Shimadzu GC-2014) equipped with an HP-5 capillary column (30 m × 0.32 mm × 0.25 µm) and a flame ionization detector. The injector and detector temperatures were set at 280 °C. The column temperature was initially maintained at 80 °C for 2 min, then heated to 260 °C at a rate of 10 °C min⁻¹, and then maintained for another 8 min. Molar yields of products from the lignin model reaction were calculated as

$$Yield\,(i) = \frac{n(i)}{n(s)} \times 100\% \tag{1}$$

where $n(i)$ and $n(s)$ are the mole of product $i$ and the initial lignin model substrate, respectively. Mass yields of aromatic monomer products from biomass reaction were calculated as

$$Yield\,(i) = \frac{m(i)}{m(b)c(l)} \times 100\% \tag{2}$$

where $m(i)$, $m(b)$, and $c(l)$ are the mass of product $i$, the mass of initial biomass, and the content of lignin determined by the NREL method,

**Table 1 | The transformation of various biomass under photothermal catalytic conditions**

| Biomass substrate | Monomers yield (wt%) | (wt%) | (wt%) |
|---|---|---|---|
| Pine sawdust | 29 | 8 | 13 |
| Platanus orientalis Linn. | 17 | 6 | 6 |
| Walnut shell | 23 | 5 | 7 |
| Reed straw | 19 | 7 | 9 |
| Wheat straw | 42 | 19 | 20 |
| Corn straw | 30 | 24 | 0 |

Reaction conditions: Biomass powders (60 mg), Pd/TiO$_2$ (10 mg), HCl (37 wt%, 10 μL), EtOH/dioxane (5:3, 1.6 mL), Kessil LED 370 nm, 140 °C, argon, 8 h. Note: The GC yields of aromatic monomers were calculated based on the weight of initial lignin in the biomass substrate.

respectively. The water-soluble depolymerized products from hemicellulose were quantified using a high performance liquid chromatography (Shimadzu LC-20AT) equipped with a C18 column. Mass yields of aliphatic monomer products from biomass reaction were calculated as

$$Yield\,(i) = \frac{m(i)}{m(b)c(h)} \times 100\% \qquad (3)$$

where $m(i)$, $m(b)$, and $c(h)$ are the mass of product $i$, mass of initial biomass, and the content of hemicellulose determined by the NREL method, respectively.

### The photocatalyst recycling procedure

After the photothermal reaction, the organic solution and solid residue were separately collected through filtration of the reaction mixture. The organic solution was directly analyzed by GC to determine the yield of lignin monomers. Besides, the compounds from hemicellulose conversion were collected via the removal of organic solvent and redissolution in water, and this aqueous solution was analyzed by HPLC to determine the yield of hemicellulose products. The solid residue after the reaction was subjected to acid-catalyzed hydrolysis which was similar to the NREL method to separate the photocatalyst from unreacted cellulose and other components. The filtrated solid from the hydrolysis mixture underwent calcination to obtain the used photocatalyst without organic impurities. The filtrated hydrolysis aqueous solution was analyzed by HPLC to determine the content of cellulose in the solid after the photothermal reaction of biomass. The used photocatalyst was subjected to the next run without further treatment.

## Data availability

The data reported in this paper including experimental details, characterization data for the catalysts and organic compounds are available in the main text and Supplementary Information files, or from the corresponding author upon request.

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

## Acknowledgements

This work was supported by the National Natural Science Foundation of China (22109139) (H.L.), the National Natural Science Foundation of China (22025206) (F.W.), the National Natural Science Foundation of China (32471809) (C.Z.), the Research Fund for High-level Talents Introduction of Nanjing Forestry University (163105107, 163105164) (C.Z.), the Dalian Innovation Support Plan for High-level Talents (2022RG13) (F.W.), and the Liaoning Revitalization Talents Program (XLYC2002012) (F.W.).

## Author contributions

H.L., C.Z., and F.W. conceived and designed the project. H.L., X.S., H.W., T.L., and X.Y. conducted the experiments. Z.Z. conducted the life cycle assessment (LCA) and techno-economic analysis (TEA). H.L. and C.Z. wrote the manuscript. All authors contributed to analyzing the data and editing the manuscript.

## Competing interests

The authors declare no competing interests.
