## [Transparent Peer Review file · Nature Communications]

Photothermal Catalytic Transfer Hydrogenolysis of Protolignin

Corresponding Author: Professor Chaofeng Zhang

Version 0:

Reviewer comments:

Reviewer #1

(Remarks to the Author)

In the present study, the authors developed a UV-induced photothermal process utilising Pd/TiO₂ for the valorisation of protolignin. A yield of phenolic monomers in a 40 wt% solution was obtained from birch sawdust. This is a noteworthy outcome. Nevertheless, I would advise against its publication in Nature Communication.

(1) The work lacks novelty. The photothermal method has been designed based on the catalytic activity of Pd for hydrogenolysis, the photocatalytic activity of TiO₂, and the in-situ protection strategy, which have all been previously reported in the literature (see the Introduction section). It is unclear how this paper contributes to the advancement of catalyst design, reaction mechanisms, or process engineering. What are the contributions of this paper to the advancement of science and technology?

(2) The paper is deficient in the study of the catalytic mechanism and the characterisation of the catalyst. In the context of photothermal reactions, it is essential to differentiate between the roles of thermal catalysis and photocatalysis. The authors of this paper have not provided any information on this topic. It would be beneficial to gain insight into the manner in which the Pd/TiO₂ catalyst interacts with UV light. It would be beneficial to gain insight into the quantum efficiencies and the effects of light intensity and wavelengths. Furthermore, it would be advantageous to understand the impact of reaction temperature on the recombination of energetic charge carriers. Additionally, providing the structure information of the Pd/TiO₂ catalyst would be invaluable for the catalytic mechanism study.

(3) In terms of practical applications, this photothermal procedure involved the use of hazardous UV, strong acid, and precious Pd. This method is not environmentally friendly nor economically viable.

Follows are some specific comments and suggestions.

(1) It would be beneficial to understand the rationale behind the specific reaction condition employed by the authors. It would be beneficial to consider the impact of varying temperatures and light sources. At 140 °C, the recombination of photogenerated electron-hole pairs is expected to be significant.

(2) In addition to TiO₂, Pd also exhibits strong absorption of UV and visible light. The use of inert carriers to support nanoparticles of Pd has been the subject of considerable research in the field of direct photocatalysis. It would be beneficial for the authors to consider the use of alternative supporting materials, such as ZrO₂ and SiO₂. Furthermore, it would be beneficial to assess the catalyst's performance under visible light conditions.

(3) Catalyst reduction was conducted at 400 °C for 4 h, which was sufficient for Pd and Pt, but may not have been adequate for Co and Ni. Ni has been widely adopted for catalysing the hydrogenolysis of lignin, due to its low cost and moderate activity. The chemical states of Ni play a pivotal role in determining the activity. Moreover, it has been demonstrated that Ni nanoparticles can function as a non-plasmonic photocatalyst. Following the appropriate processing, it is possible that Ni/TiO₂ will exhibit greater activity than Pd/TiO₂. It is worthy of investigation.

(4) It is recommended that the carbon emissions of this procedure be considered by life cycle assessment (LCA) and the economics of this procedure be considered by techno-economic analysis (TEA).

(5) For a solid-solid photoreaction, the transfer of mass and utilisation of light represent persistent challenges, particularly in the scaled-up reactors. Have the authors considered this point?

Reviewer #2

(Remarks to the Author)

This article mainly developed a photothermal catalytic process for lignin conversion using Pd/TiO₂ under UV light at 140 °C,

which can efficiently convert birch sawdust to phenolic monomers with a 40% yield in 8 hours. The 1,3-diol protection strategy is both very interesting and powerful. I think this manuscript could be accepted after minor modifications. The following suggestions could help to improve this manuscript.

1. Although photothermal catalysis has certain advantages over traditional thermal catalysis and photocatalysis, this article lacks comparison with similar work performance and does not highlight its advantages.
2. HCl may play multiple roles in the reaction system, but the exploration and description in the article are not scientific and detailed enough.
3. The analysis of biomass products is particularly complex, while the analytical methods here are not sufficiently clear, such as lacking details of the calculation formulas for productivity and yield.

Reviewer #3

(Remarks to the Author)

Li and coauthors reported direct conversion of protolignin to aromatics via the photothermal catalytic transfer hydrogenolysis process intensified by the in-situ protection strategy. The authors found that a 40 wt.% yield of phenolic monomers can be obtained when the depolymerization of birch sawdust was performed with ethanol as the hydrogen donor in 8 h, which can be ascribed to the 1,3-diol-protection between the formed C=O and the hydroxyl groups in the side chain of benzene ring. This work provided a new idea for lignin conversion. However, some issues should be carefully addressed, so I cannot recommend its publication on Nature Commun.

1. This work uses diluted HCl in the biomass conversion, however, it should be noticed that diluted HCl itself a good catalyst for the cleavage of β -O-4 in lignin. Furthermore, HCl is a catalyst for the conversion of cellulose and hemicellulose. Did the author check the performance of lignin/lignin model conversion only with HCl? Besides the monophenols, did the authors find other small molecular products? For example, furfural, HMF and ethyl levulinate? If not, what are the final states for cellulose and hemicellulose?
2. Fig. 2a, it seems that lots of reactant was formed by-products? What are the side-reactions and by-products? In general, metallic Pt has a comparable catalytic activity as Pd in many CTH reactions, however, the acetal 2 was the main products in the Pt/TiO₂ catalytic system, while the monophenols are the primary in the presence of Pd/TiO₂. Please explain this.
3. Also Fig. 2a, the increase of 3%-Pd/TiO₂ dosage resulted in an increase of guaiacol yield, but, with 6%-Pd/TiO₂, both the yields of products 3 and 4 were remarkably decreased. This seems unreasonable. The author claimed that this is because the critical synergistic effect between the semiconductor TiO₂ and loading Pd NPs. This explanation is too superficial, can the author state this clearly?
4. Line 217, "especially for the HCl, which could inhibit the oxidative cleavage of the C-C bond". This should be carefully stated. In general, a Brønsted acid favor the cleavage of C-C in oxidative depolymerization of lignin.
5. Fig. 6, the yield of monophenols in i-PrOH is significantly lower than that in PrOH. The author attributed to this to the relatively lower activity of acetone than propaldehyde. Surely, this would be a reason for this. However, the dehydrogenation capability of i-PrOH is substantially greater than PrOH, furthermore, the active H* species are crucial for the cleavage of the C-O in lignin. To be more conceivable, I suggest the author investigate this using model compounds.
6. Many logical and grammatical errors are shown, please recheck the manuscript.

Version 1:

Reviewer comments:

Reviewer #1

(Remarks to the Author)

I am pleased to see that the authors have well addressed the issues raised by the reviewers. One interesting phenomenon is that the photocurrent density of the optimal photocatalyst increased with temperature (Figure S12), which means that increasing the temperature was beneficial for the separation of photogenerated carriers. This is rare for TiO₂ photocatalysis. Can the authors explain this further? Also, I would like to see future work by the authors on Ni with this developed photothermal catalytic process.

I would recommend its publication in Nature Communications.

Reviewer #2

(Remarks to the Author)

I think the authors have fully answered my questions, and the work supported the conclusions and claims. I accept this work as it is.

Reviewer #3

(Remarks to the Author)

Surely, this work had been promoted after the revision. However, many key problems have not been solved, thus, I cannot recommend its publication on Nature Commun. This work used conventional photocatalyst of Pd/TiO₂ for the photothermal catalytic transfer hydrogenolysis of protolignin. The authors found that, 40wt% yield of phenolic monomer was obtained at 140 °C for 8 h when ethanol was used as the hydrogen donor. this monophenol yield was three times higher than the extracted 1,3-diol-protected lignin's conversion. This result more likely relates to the resource of lignin, for example, the formation of recalcitrant C-C bond in lignin separation, rather than the advantage of this cascade technique of photothermal

conversion. Furthermore, the author claimed that the C β -OAr bond before other C-O bonds, leading to a high yield of phenolic monomer products in this photothermal process. This is also just a consensus in lignin conversion. The same conclusion can also be observed by the acetalation of the side-chain of lignin with the primary alcohols. Moreover, the potential hydrogen supply of the OCH₃ in lignin should also be discussed. Thus, both the technical advance and the scientific finding are insufficient to support its publication on this high-level journal.

RESPONSE TO REVIEWER COMMENTS

Reviewer #1:

In the present study, the authors developed a UV-induced photothermal process utilising Pd/TiO₂ for the valorisation of protolignin. A yield of phenolic monomers in a 40 wt% solution was obtained from birch sawdust. This is a noteworthy outcome. Nevertheless, I would advise against its publication in Nature Communication.

(1) The work lacks novelty. The photothermal method has been designed based on the catalytic activity of Pd for hydrogenolysis, the photocatalytic activity of TiO₂, and the in-situ protection strategy, which have all been previously reported in the literature (see the Introduction section). It is unclear how this paper contributes to the advancement of catalyst design, reaction mechanisms, or process engineering. What are the contributions of this paper to the advancement of science and technology?

Response: Thank you for your comments and questions. This work targeted at developing an efficient and benign method to transform protolignin. Even though the Pd-catalyzed hydrogenolysis, the photocatalytic activity of TiO₂, and the 1,3-diol protection strategy have been well studied and reported, the photothermal method of lignin valorization was developed for the first time to overcome the difficulty or shortcoming of single photocatalytic method or single thermal-catalytic method.

As mentioned in the Introduction, thermal catalysis and photocatalysis were the two main methods for lignin hydrogenolysis. The thermal catalysis method has the advantage of fast lignin extraction and efficient hydrogenolysis of various C–O bonds, however, it needed harsh conditions (temperature above 190 °C) to generate the active hydrogen species and suffered from possible recondensation of extracted lignin and lignin fragments (Fig. 1a). The photocatalytic method has the advantage of the facile generation of reductive species under mild irradiation conditions, however, it still met the difficulty in moderate hydrogenolysis performance and the insufficient interaction between catalyst and solid lignin due to poor lignin extraction at room temperature (Fig. 1b). Based on our reasonable analyses, the photothermal method was promising to

achieve the protolignin efficient extraction-fragmentation-conversion at a relatively mild temperature with light irradiation. In the potential photothermal system (Fig. 1c), a mild but high enough temperature can ensure the release of lignin, and light irradiation over the semiconductor catalyst can induce the oxidation of sacrifice reagent by the hole (h^+) and provide active H^* species from proton and electron for the consequent hydrogenolysis of lignin linkage bonds, which can overcome the problem of efficient generation of active H^* under a mild condition for the thermal catalysis and the problem of efficient extraction of lignin from solid wood powder for the traditional photocatalysis. At the same time, the release of lignin from lignocellulosic solids can be controlled to a certain extent by lowering the temperature appropriately, which may decrease the condensation of lignin molecules in the system due to the elaborate decrease in temperature and lignin fragment concentration. Furthermore, besides the active H^* species, the reforming of primary alcohols also generates the corresponding aldehydes, which further participate in the acetalation protection of the 1,3-diol group of the *in-situ* extracted β -O-4 linkage and mediate the following selective hydrogenolysis cleavage of the C_{β} -OAr bonds to aromatic monomers. This photothermal system is not a simple combination of photocatalysis and thermal catalysis, and the synergistic effect of photocatalysis and thermal catalysis is crucial to the prior cleavage of the C_{β} -OAr bond, avoiding the formation of an inert benzylic dehydroxylated byproduct to prevent the C_{β} -OAr bond cleavage.

We have learned from previous system design strategies but organically and creatively combined the corresponding systems based on the crucial challenge analysis. Therefore, the overall presentation of this paper, a pioneering research for the lignin photothermal transformation, can be considered as a new research direction of lignin catalytic conversion. At the same time, as the comparison and analysis of the yield of aromatic monomers between this photothermal method and traditional photocatalytic or thermal catalytic methods (listed in the revised manuscript and supporting information, Table 1, Fig. 8, and Table S1), lignin photothermal catalytic conversion and utilization is a new research hotspot and a method with application amplification potential for process engineering. We added new discussions and explorations for the

catalyst design/characterization and reaction mechanism based on your suggestions.

In brief, the novelty of this work is to develop photothermal catalysis for transfer hydrogenolysis of lignin to overcome the shortcomings of single thermal catalysis or photocatalysis. In addition, we revised Figure 1 and added the following discussion in the manuscript and Supporting Information.

‘Photothermal catalysis has emerged as a promising strategy to combine the advantages of single thermal catalysis with that of photocatalysis.’

‘Compared to the monomer yields from reported works focusing on the lignin and protolignin valorization via transfer hydrogenolysis, this photothermal system delivered an ideal monomer yield that was close to the theoretical value but under a milder condition (Table S1, for more information).’

‘In brief, we realized the efficient protolignin conversion to aromatics via the photothermal catalytic transfer hydrogenolysis intensified by the in-situ extraction-protection strategy to overcome the shortcomings of single thermal catalysis or photocatalysis.’

(a) Hydrogen-free reductive (thermal) catalytic fractionation for lignin-first biorefining

😊 Fast in-situ lignin extraction; High monomers yield. 😞 Harsh conditions; Possible lignin recondensation.

(b) Photocatalytic self-transfer hydrogenolysis of lignin in lignocellulose

😊 Mild conditions; No need of external hydrogen donor. 😞 Difficult lignin extraction; Low monomers yield and selectivity.

(c) This work: Photothermal catalytic transfer hydrogenolysis of protolignin via the *in-situ* protection

😊 Solar-driven photothermal catalysis; Relative mild conditions; Controllable in-situ lignin extraction; High monomers yield.

Fig. 1 | A comparison of developed thermal catalytic methods, photocatalytic methods, and our proposed photothermal catalytic method for transfer hydrogenolysis of protolignin in lignocellulose.

(2) The paper is deficient in the study of the catalytic mechanism and the characterisation of the catalyst. In the context of photothermal reactions, it is essential to differentiate between the roles of thermal catalysis and photocatalysis.

The authors of this paper have not provided any information on this topic. It would be beneficial to gain insight into the manner in which the Pd/TiO₂ catalyst interacts with UV light. It would be beneficial to gain insight into the quantum efficiencies and the effects of light intensity and wavelengths. Furthermore, it would be advantageous to understand the impact of reaction temperature on the recombination of energetic charge carriers. Additionally, providing the structure information of the Pd/TiO₂ catalyst would be invaluable for the catalytic mechanism study.

Response: Thank you for your comments and suggestions. We divided this comment into six suggestions.

(A) The roles of thermal catalysis and photocatalysis.

As reported in the reviews on photothermal catalysis (Joule 2024, 8, 312–333; doi:10.1021/acsnm.4c00598), there are mainly four types of photothermal catalysis: thermal-assisted photocatalysis, photoassisted thermal catalysis, photo-driven thermal catalysis, and photo-thermo cascade catalysis. Our developed photothermal catalytic transfer hydrogenation could be identified as photo-thermo cascade catalysis based on the following facts. (I) No obvious conversion of the lignin model was observed with heating but without light irradiation (Fig. 3b). Therefore, photoassisted thermal catalysis and photo-driven thermal catalysis can be excluded from the current system. (II) The hydrogenolysis selectivity of the C_β-OAr bond rather than the conversion of the substrate without C_β-OAr bond cleavage was dramatically improved by elevating the reaction temperature under light irradiation (Fig. 3a). Therefore, thermal-assisted photocatalysis can be also excluded. (III) The efficient formation of 1,3-diol protected intermediate (acetal **2**) which involved the photocatalytic generation of acetaldehyde and thermal catalytic condensation between the 1,3-diol model with acetaldehyde can be definitely assigned as photo-thermo cascade catalysis. This process was vital in the transformation of protolignin because the in-situ extracted lignin tended to undergo recondensation via the formation of C–C bonds, but the condensation can be restrained by the in-situ protection step (acetalization) during this photothermal system.

(B) The interaction between Pd/TiO₂ with UV light.

We supplemented the absorption, photocurrent, and photoluminescence in the revised manuscript. The loading of Pd induced the absorption at the visible light region (Fig. S8, a). Both 3 wt%-Pd/TiO₂ and 6 wt%-Pd/TiO₂ showed weak fluorescence compared with TiO₂ (Fig. S8, b), suggesting that the charge-separation efficiency was obviously improved because of the loaded Pd particles. The 3 wt%-Pd/TiO₂ showed a stronger photocurrent than TiO₂, which was consistent with the fluorescence result (Fig. S8, c). However, the 6 wt%-Pd/TiO₂ showed a weaker photocurrent than TiO₂. The loading of excess Pd particles on the surface may affect the light absorption of TiO₂ at 370 nm which induced the weak photocurrent. The above discussion and Fig. S8 were added to the supporting information.

The following discussion was added to the manuscript. *'The loading of Pd particles induced the visible light absorption and promoted the charge-carriers separation (Fig. S8)'*

Fig. S8 | The photo-responsive properties of photocatalysts. (a) UV-vis absorption. (b) fluorescence spectra ($\lambda_{\text{EX}} = 300 \text{ nm}$). (c) Photocurrent test (under 370 nm LED irradiation).

(C) The quantum efficiencies

The apparent quantum efficiency (ζ) of transfer hydrogenation based on the transformation of the lignin model was calculated.

Fig. S10 | The proposed elementary steps involved in the transfer hydrogenolysis

Fig. S11 | The kinetic data used for determining the initial rate of model 1

The transformation data at the initial stage (10 min) was used to calculate the kinetic data.

$$\text{rate}(\text{H}^*) = \text{rate}(\mathbf{3}) \times 6 = n(\mathbf{3})/t \times 6 = (10 \text{ mg})/(272 \text{ mg/mmol}) \times 0.208/(0.1 \text{ h}) \times 6 = 0.459 \text{ mmol/h}$$

$$I_0(370 \text{ nm}) = P/E(370 \text{ nm})/N_A = \text{Avg. Intensity} \times \text{Area} \times \lambda/hc/N_A = (137 \text{ mW/cm}^2) \times (1.8 \text{ cm}^2)/(6.626 \times 10^{-34} \text{ J}\cdot\text{s})/(299792458 \text{ m/s}) \times (370 \times 10^{-9} \text{ m})/(6.022 \times 10^{23} \text{ mol}^{-1}) = 7.63 \times 10^{-7} \text{ mol/s} = 2.74 \text{ mmol/h}$$

$$\zeta = \text{rate}/I_0 \times 100\% = 16.7\%$$

These calculation details were added to the supporting information. The following discussion was added to the manuscript. ‘*The apparent quantum efficiency was 16.7% based on the proposed elementary reaction steps (Fig. S10) and kinetic data obtained at the initial stage (Fig. S11).*’

(D) The effects of light intensity and wavelengths on the photothermal reaction

We supplemented more tests on the effect of light sources according to your suggestion. The decreasing of the input powder of the 370 LED lamp obviously decreased the yield of guaiacol and *n*-propylbenzene, which suggested that sufficient light irradiation was vital for the efficient transfer hydrogenolysis of β -O-4 linkage. No reaction occurred under visible light irradiation (427 and 456 nm). Even the introduction of Pd particles onto TiO₂ induced the visible light absorption of the whole catalyst, however, the visible light irradiation could not excite TiO₂. The oxidation of the hydrogen donor was the initial step for the whole transfer hydrogenolysis process, and only excited TiO₂ could efficiently initiate the oxidation of the hydrogen donor at the valence band. Table S3 was added to the supporting information and the following discussion was added in the manuscript. ‘*Initially, this transformation diminished with decreasing of 370 nm light intensity and failed after switching the UV light to visible light (Table S3), which suggests the UV-triggered photocatalysis was essential for the whole transformation.*’

Table S3 | The effect of the light source condition on photothermal catalytic transfer hydrogenolysis of lignin model

Entry	Wavelength (nm)	Input power (W)	Yield of 3 (%)	Yield of 4 (%)	Yield of 5 (%)
1	370	0	0	0	0
2	370	10	56	18	22
3	370	20	62	45	16
4	370	40	78	67	20
5	427	40	0	0	0
6	456	40	0	0	0

Conditions: substrate **1** (10 mg), 3%-Pd/TiO₂ (10 mg), EtOH (1 mL), HCl (37%, 10 μ L), Kessil LED (370/427/456 nm), 0-40 W, 140 °C, argon, 2 h.

(E) The impact of reaction temperature on the recombination of energetic charge carriers

The transient photocurrent response is a common method to determine the recombination of charge carriers. Here the transient photocurrent response under different temperatures was used to reveal the impact of temperature on the recombination of photogenerated charge carriers. It was observed that the photocurrent density of the optimal photocatalyst increased with the temperature. Therefore, the elevating of temperature was beneficial to the separation of photo-generated charge carriers. We added this result to the following discussion in the manuscript. *‘For the promotion effect of the temperature on the photothermal system, further photocurrent characterization of the Pd/TiO₂ catalyst at 30, 94, and 140 °C indicated the photocurrent slightly increased with the temperature, which suggested that heating favored the separation of photo-generated charge carriers (Fig. S12).’*

Fig. S12 | The photocurrent test of 3%-Pd/TiO₂ under 370 nm LED irradiation at different temperatures

(F) More characterization of the catalyst

We supplemented more structure and morphology characterizations on the photocatalyst. X-ray diffraction (XRD) patterns indicated that TiO₂ support contained anatase as the major phase and rutile as the minor phase, which was in accordance with reported Degussa P25. The Pd (111) peak was also observed suggesting the formation of Pd particles on TiO₂. The loading of Pd slightly decreased the Brunauer-Emmett-

Teller (BET) surface area of catalyst, from 63.9 to 46.4-47.7 m²/g. The amount of Pd loading obviously affected the size of Pd particles according to the High Resolution Transmission Electron Microscope (HRTEM) images. The mean size of Pd particles in 3%-Pd/TiO₂ was 5.1 nm, while that in 6%-Pd/TiO₂ was 8.1 nm. The larger size of metal particle will induce a low percentage of surface atom. Therefore, the higher loading amount of Pd may not induce more active metal sites. The Energy Dispersive Spectroscopy (EDS) mapping indicated that Pd particles were evenly dispersed on the TiO₂ without formation of agglomerate. X-ray photoelectron spectroscopy (XPS, Figure 7d) indicated the surface Pd was composed of 92% Pd⁰ species and 8% Pd²⁺ species on the fresh catalyst. We added following discussions in the revised manuscript. *‘The structure and morphology characterizations (Fig. S5-S7 and Table S2) of the optimal catalyst indicated that the metallic Pd nanoparticles with a mean size of 5.1 nm were evenly dispersed on the surface of TiO₂ particles’*

Fig. S5 | The XRD pattern of prepared 3%-Pd/TiO₂

Table S2 | The BET analysis of photocatalysts

Sample	BET surface area (m ² /g)
TiO ₂	63.9
3%-Pd/TiO ₂	47.7
6%-Pd/TiO ₂	46.4

Fig. S6 | The SEM (a) and TEM/HRTEM images (b, c) of 3%-Pd/TiO₂, and SEM (d) and TEM/HRTEM images (e, f) of 6%-Pd/TiO₂

Fig. S7 | The TEM-EDS mapping images of 3%-Pd/TiO₂

(3) In terms of practical applications, this photothermal procedure involved the use of hazardous UV, strong acid, and precious Pd. This method is not environmentally friendly nor economically viable.

Response: Thank you for your comments. Although UV light was used in the model test and mechanism study, solar light contains the UV part of light and we have proved that solar light can act as the light source and heat source for the transformation of several biomass. The utilization of solar light makes our method environmentally friendly and economically viable. In addition, ultraviolet light is a reasonable excitation light source and has been widely used in organic synthesis, biomass conversion, and small molecule conversion, such as water decomposition. High energy excitation light is required because the band gap of some efficient and cheap semiconductor photocatalysis is too wide (TiO₂ is a typical example).

The major contribution of this work is to develop photothermal catalysis for transfer hydrogenolysis of lignin to overcome the shortcomings of single thermal catalysis or photocatalysis. The photothermal system involves reforming the alcohol and the selective hydrogenolysis of the lignin substrate with selective bond transformation, which is the expertise area of precious metals like Pd, not the single semiconductor. In addition, Pd and precious metals are widely used in biomass conversion. During this research, we have proved that the photocatalyst Pd/TiO₂ can be recycled and reused four times without obvious deactivation. Therefore, the cycle stability of heterogeneous Pd/TiO₂ can remarkably decrease the cost of catalyst in our system. Therefore, the use of the Pd is not a restrictive factor. The utilized photocatalyst and acid catalyst were common catalysts for hydrogenation and lignin extraction, respectively. The addition of a catalytic amount of HCl was vital for in situ lignin extraction and enhanced selectivity of C–O bond cleavage. Furthermore, HCl is a cheap and powerful acid catalyst and the possible corrosion or environmental issues can be further controlled by further investigating the potential recyclable solid acid or grafting the organic acid molecules on the heterogeneous photocatalyst. In addition, to reduce the use of H₂SO₄ during the catalyst recycling from the mixture of catalyst and residue of lignocellulose,

we are studying the potential catalyst with an inert magnetic core for the potential application amplification.

The following discussion was added to the revised manuscript. ‘*The cycle stability of Pd/TiO₂ can decrease the cost of precious metal in further large-scale biomass refinery.*’ ‘*The utilization of solar light makes our method environmentally friendly and economically viable.*’

Follows are some specific comments and suggestions.

(1) It would be beneficial to understand the rationale behind the specific reaction condition employed by the authors. It would be beneficial to consider the impact of varying temperatures and light sources. At 140 °C, the recombination of photogenerated electron-hole pairs is expected to be significant.

Response: Thank you for your suggestions. Detailed investigations on reaction conditions including solvent, temperatures, and light sources were performed. Besides ethanol, other alcohols including methanol, 1-PrOH, 2-PrOH, 1-BuOH, and 2-BuOH were also utilized as the hydrogen donor and solvent for the photothermal transformation. These alcohols delivered a slightly lower yield of hydrogenolysis products which demonstrated that ethanol was the best solvent and hydrogen donor.

The synergistic effect between photocatalysis and thermal catalysis in this system played a critical role in the reaction performance (Fig. 3). The Pd/TiO₂ system tended to catalyze the hydrogenolysis of C_α-OH into C_αH₂ (5, 67% yield) rather than the C_β-OAr ether bond cleavage under light irradiation at room temperature (Fig. 3a), which was consistent with our previous results that the benzylic C_α-O bond was reactive on Pd/TiO₂ at room temperature (Green Chem., 2020, 22, 3802-3808. ChemCatChem, 2022, 14, e202200120). However, with the assistance of heat by raising the temperature to 140 °C (Fig. 3a), the β-O-4 diol conversion over the Pd/TiO₂ system tended to catalyze the C_β-OAr ether bond cleavage and provided *n*-propylbenzene and guaiacol as the monomer products, which involved the simultaneous hydrogenolysis of C_α-OH, C_β-OAr, and C_γ-OH.

The decreasing of the input power of the 370 nm LED lamp obviously decreased

the yield of guaiacol and *n*-propylbenzene, which suggested that sufficient light irradiation was vital for the efficient transfer hydrogenolysis of β -O-4 linkage. No reaction occurred under visible light irradiation (427 and 456 nm). Even the introduction of Pd particles onto TiO₂ induced the visible light absorption of the whole catalyst, however, the visible light irradiation could not excite TiO₂. The oxidation of the hydrogen donor was the initial step for the whole transfer hydrogenolysis process, and only excited TiO₂ could efficiently initiate the oxidation of the hydrogen donor at the valence band.

The transient photocurrent response is a common method to determine the recombination of charge carriers. Herein the transient photocurrent response under different temperatures was used to reveal the impact of temperature on the recombination of photogenerated charge carriers. It was observed that the photocurrent density of the optimal photocatalyst increased with the temperature. Therefore, the elevating of temperature was beneficial to the separation of photo-generated charge carriers.

Based on these, the following discussions were added to the manuscript. *‘Besides ethanol, other simple aliphatic alcohols also can be utilized as the hydrogen donor and solvent, delivering slightly lower yields of hydrogenolysis products (Fig. S9).’*

‘Initially, this transformation diminished with decreasing of 370 nm light intensity and failed after switching the UV light to visible light (Table S3), which suggests the UV-triggered photocatalysis was essential for the whole transformation.’

‘the hydrogenolysis selectivity of the C β -OAr bond rather than the conversion of the substrate without linkage cleavage was dramatically improved by increasing the reaction temperature under light irradiation (Fig. 3a).’

‘For the promotion effect of the temperature on the photothermal system, further photocurrent characterization of the Pd/TiO₂ catalyst at 30, 94, and 140 °C indicated the photocurrent slightly increased with the temperature, which suggested that heating favored the separation of photo-generated charge carriers (Fig. S12).’

Fig. S9 | The photothermal catalytic transfer hydrogenolysis of lignin model in different alcohols. Reaction conditions: substrate 1 (10 mg), 3%-Pd/TiO₂ (10 mg), solvent (1 mL), HCl (37%, 10 μ L), Kessil LED (370 nm), 140 °C, argon, 2 h.

Fig. 3a | The effect of thermal heating on the photocatalytic hydrogenolysis pathways. Reaction conditions: substrate 1 (10 mg), 3%-Pd/TiO₂ (10 mg), EtOH (1 mL), HCl (37%, 10 μ L), Kessil LED (370 nm), 30-160 °C, argon, 2 h.

Table S3 | The effect of the light source on photothermal catalytic transfer hydrogenolysis of lignin model

Entry	Wavelength (nm)	Input power (W)	Yield of 3 (%)	Yield of 4 (%)	Yield of 5 (%)
1	370	0	0	0	0
2	370	10	56	18	22
3	370	20	62	45	16
4	370	40	78	67	20
5	427	40	0	0	0
6	456	40	0	0	0

Conditions: substrate **1** (10 mg), 3%-Pd/TiO₂ (10 mg), EtOH (1 mL), HCl (37%, 10 μ L), Kessil LED (370/427/456 nm), 0-40 W, 140 $^{\circ}$ C, argon, 2 h.

Fig. S12 | The photocurrent test of 3%-Pd/TiO₂ under 370 nm LED irradiation at different temperatures

(2) In addition to TiO₂, Pd also exhibits strong absorption of UV and visible light. The use of inert carriers to support nanoparticles of Pd has been the subject of considerable research in the field of direct photocatalysis. It would be beneficial for the authors to consider the use of alternative supporting materials, such as ZrO₂ and SiO₂. Furthermore, it would be beneficial to assess the catalyst's performance under visible light conditions.

Response: Thank you for your suggestions. The above investigation on the effect of light source has demonstrated that no catalytic transfer hydrogenolysis activity was

shown by using Pd/TiO₂ under visible light irradiation. Therefore, the Pd particles under visible light irradiation could not trigger the reaction. The Pd particles loading on inert support such as ZrO₂ and SiO₂ also were invalid for the reaction under UV and visible light irradiation (Table S4, entries 6-9), which further precludes the possible photocatalytic role of Pd particles. The photocatalytic hydrogenolysis activity was only observed when semiconductors were used in the carriers (Table S4, entries 1-5, 10, and 11). Besides the P25 TiO₂, the anatase, rutile, CeO₂, Nb₂O₅, ZnO, and g-C₃N₄ could also deliver the hydrogenolysis products with loading of the same amount of Pd particles under light irradiation.

Table S4 | The photothermal catalytic test of Pd-loaded catalysts under UV and visible light irradiation.

Entry	Catalyst	Light wavelength (nm)	Yield of 2 (%)	Yield of 3 (%)	Yield of 4 (%)	Yield of 5 (%)
1	Pd/anatase	370	3	44	46	13
2	Pd/rutile	370	33	4	8	6
3	Pd/CeO ₂	370	0	6	25	0
4	Pd/Nb ₂ O ₅	370	7	5	17	0
5	Pd/ZnO	370	0	0	31	0
6	Pd/ZrO ₂	370	0	0	0	0
7	Pd/ZrO ₂	427	0	0	0	0
8	Pd/SiO ₂	370	0	0	0	0
9	Pd/SiO ₂	427	0	0	0	0
10	Pd/g-C ₃ N ₄	370	12	42	45	0
11	Pd/g-C ₃ N ₄	427	9	42	40	24

Conditions: substrate **1** (10 mg), catalyst (3 wt% Pd, 10 mg), EtOH (1 mL), HCl (37%, 10 μ L), Kessil LED (370/427 nm), 140 °C, argon, 2 h.

The following discussions were added to the manuscript. ‘*Although the loading of Pd particles could induce visible light absorption and promote the charge-carriers separation (Fig. S8), no visible light-responsive catalytic activity was observed for Pd/TiO₂ (Table S3). Meanwhile, the photocatalytic transfer hydrogenolysis occurred only when Pd particles were loaded on semi-conductors (ZnO, Nb₂O₅, CeO₂, and C₃N₄) rather than inert carriers (SiO₂ and ZrO₂) under UV or visible light irradiation (Table S4).*’

(3) Catalyst reduction was conducted at 400 °C for 4 h, which was sufficient for Pd and Pt, but may not have been adequate for Co and Ni. Ni has been widely adopted for catalysing the hydrogenolysis of lignin, due to its low cost and moderate activity. The chemical states of Ni play a pivotal role in determining the activity. Moreover, it has been demonstrated that Ni nanoparticles can function as a non-plasmonic photocatalyst. Following the appropriate processing, it is possible that Ni/TiO₂ will exhibit greater activity than Pd/TiO₂. It is worthy of investigation.

Response: Thank you for your suggestion. It is appealing to achieve hydrogenolysis using non-precious metals rather than Pd or Pt. The Ni/TiO₂ samples reduced under different temperatures (400-600 °C) were prepared and tested for this transformation. The higher calcination temperature than 600 °C was abandoned because it will induce the TiO₂ crystal phase transition from anatase to rutile which will dramatically affect the photocatalytic performance. Unfortunately, higher reduction temperatures could not endow Ni/TiO₂ with the hydrogenolysis activity under current mild conditions and only **2** was obtained as the product. Normally a higher reaction temperature was needed for non-precious metals catalyzed hydrogenation than precious metals. The mild photothermal conditions used in this work limited the utilization of Ni as the hydrogenation sites.

Table S5 | The photothermal catalytic test of Ni-loaded catalysts

Entry	Catalyst	Yield of 2 (%)	Yield of 3 (%)	Yield of 4 (%)	Yield of 5 (%)
1	Ni/TiO ₂ -400	57	0	0	0
2	Ni/TiO ₂ -500	40	0	0	0
3	Ni/TiO ₂ -600	46	0	0	0

Conditions: substrate **1** (10 mg), catalyst (3 wt% Ni, 10 mg), EtOH (1 mL), HCl (37%, 10 μL), Kessil LED (370 nm), 140 °C, argon, 2 h. The catalysts reduced with hydrogen under 400, 500, and 600 °C were denoted as Ni/TiO₂-400, Ni/TiO₂-500, and Ni/TiO₂-600, respectively.

(4) It is recommended that the carbon emissions of this procedure be considered by life cycle assessment (LCA) and the economics of this procedure be considered by techno-economic analysis (TEA).

Response: The photothermal hydrogenolysis of wheat straw to 2-methoxy-4-propylphenol under sunlight was analyzed via LCA and TEA. The following results were added to the manuscript.

‘Techno-economic analysis and life cycle assessment (LCA)

An industrial-scale process flow has been designed and simulated to evaluate the practical potential of 4-propylguaiacol production from the solar-driven photothermal conversion of wheat straw, which featured the extensive source feedstock and high selectivity of a single product. The model employed wheat straw as feedstock with 100,000 metric tons per annual (t/a), which approximates 2800 t/a of 4-propylguaiacol yield. With feedstock pretreatment, photothermal reaction, consumable recovery, 4-propylguaiacol purification, cellulose hydrolysis, and photocatalyst recovery units (Fig. S16), the model was performed through steady-state simulation using Aspen plus to obtain material and energy inventories according to current reaction conditions and parameters (Table S9). Techno-economic analysis and life cycle carbon emissions were subsequently carried out to identify opportunities for industrialization. The results reveal that the total production cost (TPC) of 4-propylguaiacol can be lower at 44354 CNY/t, and consumables and depreciation costs are two constraints due to the small scale designed based on the biomass supply and product demand. Life cycle greenhouse gases (GHG) emissions is 8.70 tCO₂eq/t 4-propylguaiacol, of which wheat straw accounts for the largest contribution of 85.28%. The life cycle GHG emissions of wheat straw approximates to 170 kg CO₂eq/t due to plant, harvest, and transport. High wheat straw input causes more GHG emissions than the carbon credit of bio-derived chemicals, thus a higher conversion ratio is attractive to facilitate this work. The details can be seen in the Supplementary Information.’

The detailed of TEA and LCA were provided in the supporting information.

‘ Process design details (TEA and LCA)

The process design is divided into two major sections: solar-driven photothermal reaction of protolignin, and cellulose hydrolysis and photocatalyst recovery. The front process is further split into feedstock pretreatment, photothermal reaction, gas and consumable recovery, residue separation, and 4-propylguaiacol purification units, while cellulose hydrolysis and photocatalyst recovery include unreacted cellulose hydrolysis and photocatalyst recovery units. Process design details are depicted in the following section.

The solar-driven photothermal reaction of protolignin

The feedstock is wheat straw that consists of cellulose, hemicellulose, lignin, ash, and water, as shown in Table S1. The milled wheat straw, together with process water, photocatalyst, hydrochloric acid, ethanol, dioxane, and argon flow, was heated for 50 °C and then fed into a photoreactor, where the catalytic reaction occurs under solar-driven photothermal conditions for a certain time. The hemicellulose and lignin were broken free from wheat straw and further converted into 4-propylguaiacol, monomers, lactic acid, arabinose, xylose, acetic acid, levulinic acid, polysaccharide. The cellulose and ash were sunk into slurry. Thus, the outlet mixture consists of the products derived from hemicellulose and lignin, slurry from cellulose and ash, together with catalyst, solvent, and argon atmosphere.

The separation sequence was arranged to obtain 4-propylguaiacol with a purity of 99.5wt% as well as co-products recovery. At first, the mixture products were fed into a gas-liquid separator and filter-press to remove gas- and solid-phase products. The gas was mainly argon used to be recycled as feedstock, while the solid contains cellulose, ash, and photocatalyst that requires further separation in the unit of cellulose hydrolysis and photocatalyst recovery. The residual mixture was fed into a flash tank at 200 °C to evaporate liquids with low boiling points, such as ethanol, dioxane, and water, which are solvents of photoreaction and recycled. The rest of the liquids with high boiling points were fed into a decanter to remove hydrophilic components through

solubility in water, and then the 4-propylguaiacol and some dimers were obtained. Finally, a distillation column was assigned to recover the 4-propylguaiacol with designated purity.

Table S8 | Composition of wheat straw used for TEA and LCA analysis

Component	Content	Component in simulation model
Cellulose	33.0 wt%	Cellulose
Hemicellulose	23.2 wt%	DI-Xylose
Lignin	13.0 wt%	Vanillin
Soluble components	18.8 wt%	Assumed to be glucose
Ash	12.0 wt%	Unconventional component

Slurry concentration and photocatalyst recovery

The slurry obtained from filter contains cellulose, ash, and photocatalyst, during which photocatalyst is required to recover due to its high cost. In addition, cellulose hydrolysis can produce costly glucose but suffers from complex processes, thus two scenarios with or not a cellulose hydrolysis process are arranged to identify higher economic performance and lower carbon emissions. This part is to reach cellulose hydrolysis and photocatalyst recovery.

Scenario 1: *The cellulose hydrolysis occurs under 130 °C with the participation of sulfuric acid. Specifically, slurry derived from the filter was successively fed into a washer and digester. The washer was full of sulfuric acid at 45 °C while the digester was heated to 130 °C. During this process, nearly all cellulose was broken down into glucose. Then the mixture delivered from the digester cooled, and the aqueous solution, such as H₂SO₄, glucose, etc., was removed. The rest entered the incinerator to burn*

solid organics and recover the photocatalyst, meanwhile, the steam generated supplied for whole heat network.

Scenario 2: Slurry derived from the filter was delivered into the incinerator to burn solid organics to recover photocatalyst and generate steam.

According to previous work and the above parameters, the overall process design was simulated by Aspen plus to estimate detailed material and energy flow. The assumed input mass of wheat straw is 100,000 metric tons per annual (t/a). The output amount of 4-propylguaiacol will be 2800 t/a, which is a receivability scale. The process flow model is shown in Fig. S16. The components of the key stream are shown in Table S9. Note that the steam derived from the incinerator is sufficient for the whole plant in Scenario 2.

Fig. S16 | Process flow model of Scenario 1(a) and Scenario 2(b)

***Table S9 | Key stream for process flow model***

| | *Feed* | *Makeup* | *Raw product* | *Argon** | *Solvent** | *Slurry* | *Water* | *Product* | *Acid* | *Steam* | *Hydrolysis* | *Syrup* | *Catalyst* |
|---|---|---|---|---|---|---|---|---|---|---|---|---|---|
| *Mass* | *10* | *0.10* | *15.95* | *4.06* | *0.16* | *8.09* | *333.33* | *0.28* | *166.67* | *2833.33* | *3008.09* | *3001* | *1.67* |
| *Mass fraction/%* | | | | | | | | | | | | | |
| *Wheat straw* | *100* | | | | | | | | | | | | |
| *Water* | | *0.33* | *0.01* | *0.02* | *0.36* | | *100* | | *28.00* | *100* | *95.74* | *95.94* | |
| *Dioxane* | | *35.98* | *0.63* | *0.56* | *25.05* | | | | | | | | |
| *Ethanol* | | *24.55* | *0.82* | *1.00* | *40.58* | | | | | | | | |
| *Argon* | | *39.15* | *25.07* | *98.40* | *1.06* | | | | | | | | |
| *Lactic acid* | | | *7.01* | | *23.19* | | | | | | | | |
| *Acetic acid* | | | *0.05* | *0.02* | *2.07* | | | | | | | | |

Levulinic acid	0.04	0.07			
Xylose	2.92	0.84			
Monomers	6.39	0.07	0.42		
Glucose				0.06	0.06
Arabinose	4.53	1.30			
Cellulose	9.77		19.26		
4-Propylguaiacol	1.81	5.40	99.58		
Polysaccharide	12.08		23.81	0.06	
Slurry	18.43		36.34	0.10	
Catalyst	10.45		20.59		100
Sulfuric acid				72.00	3.99
					4.00

**1% of the flow was used as the purge gas to maintain the elemental balance of the system.*

Techno-economic analysis (TEA)

The total production cost (TPC) of 4-propylguaiaicol was estimated to evaluate the economic feasibility of this work. The assumed annual plant operating time is 8000 h and the total project life is taken to be 20 years. Depreciation cost is estimated with a straight-line depreciation method.

The total capital investment (TCI) consists of fixed capital investment (FCI) and working capital. The FCI of each equipment is estimated according to its inside battery limits (ISBL), outside battery limits (OSBL), and indirect costs (IDC). The ISBL was determined by Aspen Process Economic Analyzer, while the OSBL and IDC were estimated to ratios of the ISBL. Working capital is used to maintain plant operation, which was estimated as 10% of the FCI²⁹.

The TPC is estimated by raw materials cost, consumables cost, utilities cost, operating and maintenance cost, plant overhead cost, administrative cost, and byproduct revenue. The raw materials cost, consumables cost, utilities cost, and byproducts revenue were calculated by multiplying production inventories (shown in Table S9) with market prices (shown in Table S10). Other costs were estimated as coefficients of the FCI and operating labor costs. The operating labor cost was estimated according to operator salaries, plant scale, and equipment characteristics²⁹.

Table S10 | Key parameters for techno-economic analysis

Parameter	Remarks
TCI	(v)+(vi)
Inside battery limits (ISBL)	(i) Installed equipment cost
Outside battery limits (OSBL)	(ii) 40% of (i)
Direct costs	(iii) (i) + (ii)
Indirect costs (IDC)	(iv) 60% of (iii)
Fixed capital investment (FCI)	(v) (iii) + (iv)
Working capital	(vi) 10% of (v)
TPC	(1)+(2)+(3)+(4)+(5)+(6)+(7)+(8)–(9)
Raw materials cost	(1) Wheat straw: 50 CNY/t
Consumables cost	(2) Hydrochloric acid: 82 CNY/t Sulfuric acid: 400 CNY/t Ethanol: 6000 CNY/t Dioxane: 5000 CNY/t Argon: 1000 CNY/t
Utilities cost	(3) Steam: 200 CNY/t Process water: 15 CNY/t
Operating and maintenance cost	(4) (4.1)+(4.2)+(4.3)+(4.4)+(4.5)

Operating labor	(4.1)	100000 CNY/operator year, 58 operators
Direct supervisory and clerical labor	(4.2)	20% of (4.1)
Maintenance and repairs	(4.3)	2% of (v)
Operating supplies	(4.4)	0.8% of (v)
Laboratory charge	(4.5)	15% of (4.1)
Depreciation cost	(5)	Lifetime: 20 years, salvage value: 4%
Plant overhead cost	(6)	60% of ((4.1)+(4.2)+(4.3))
Administrative cost	(7)	2% of TPC
Distribution and selling cost	(8)	2% of TPC
Byproduct revenue	(9)	Glucose: 2800 CNY/t dry

Techno-economic analysis results

The TPC of 4-propylguaiacol in this work was estimated to be 44354 CNY/t in Scenario 2, which is lower than that (187877 CNY/t) in Scenario 1. The high TPC of 4-propylguaiacol in Scenario 1 is attributable to huge steam input in the cellulose hydrolysis process, which implies the deprecated process of cellulose hydrolysis in this work. Apart from the cellulose hydrolysis process, the processes are parallel in Scenario 1 and 2. The main contribution accounts for consumables cost and depreciation cost. The reason might be that small scale (2800 t 4-propylguaiacol per annum) designed based on the biomass supply and product demand restrict the usage of advanced but costly facilities to recover consumables. The detailed breakdown of the TPC of 4-propylguaiacol in Scenario 2 is shown in Fig. S17.

Fig. S17 | The detailed breakdown of total production cost (TPC) of 4-propylguaiacol in Scenario 2

Life cycle assessment (LCA)

Life cycle assessment (LCA) is used to evaluate greenhouse gas (GHG) emissions of this work. LCA is effective in evaluating environmental benefits, which is defined as the “compilation and evaluation of the inputs, outputs and potential environmental impacts of a product system throughout its life cycle”.³⁰ The functional unit is defined as 1 metric ton of 4-propylguaiacol. The system boundary covers GHG emissions for the wheat straw collection, transport, production of consumables (hydrochloric acid, ethanol, dioxane, argon, and sulfuric acid), conversion process simulated above, and all upstream of required materials and utilities. The detailed system boundary of 4-propylguaiacol is depicted in Fig. S18.

Fig. S18 | System boundary of this work

The GHGs involve CO_2 , CH_4 , and N_2O and were estimated in units of $kg\ CO_2$ equivalent according to a 100-year-time horizon³¹, as Eq(S1).

$$GHG = E_{CO_2} + 25E_{CH_4} + 298E_{N_2O} - 44/12C_{prod} \quad (S1)$$

where E_{CO_2} , E_{CH_4} and E_{N_2O} represent emissions of CO_2 , CH_4 and N_2O , respectively, kg ; C_{prod} represents carbon sequestration in products, kg .

REET 2020 software was taken to develop the model and link units³². The inventory of transformation from wheat straw to 4-propylguaiacol was shown in Supplementary Table S2 as mentioned above. Collection and transport of wheat straw, electricity, steam, hydrochloric acid, ethanol, dioxane, argon, sulfuric acid, and other background inventory, were taken from the inherent database of REET software. Life cycle GHG emissions were obtained by incorporating estimated material and energy into REET 2020 finally.

Life cycle assessment results

The life cycle GHG emissions are 103.81 and 8.70 tCO_2eq/t 4-propylguaiacol in Scenario 1 and Scenario 2 (Fig. S16), respectively. The high GHG emissions are responsible for steam consumption (89.33%) in Scenario 1. The contribution of each factor to GHG emissions in Scenario 2 is shown in Fig. S19. It can be seen that the wheat straw accounts for the largest contribution, implying a higher conversion ratio is required.

Fig. S19 | The detailed breakdown of life cycle GHG emissions of 4-propylguaiacol in Scenario 2

(5) For a solid-solid photoreaction, the transfer of mass and utilization of light represent persistent challenges, particularly in scaled-up reactors. Have the authors considered this point?

Response: Thank you for your comments and questions. The dioxane was added as the second solvent for the transformation of biomass. During the reaction, the lignin fragments were in situ extracted from solid lignocellulose by dioxane-alcohol mixture and dissolved in the liquid phase. Then the heterogeneous photocatalyst interacted with the soluble lignin fragments to generate the aromatic monomer products. Therefore, this system is not a solid-solid photoreaction but a liquid-solid photoreaction (Energy Environ. Sci. 6, 994-1007 (2013)).

We agree with you that the developed one-pot reaction system in the current work will meet the challenges in mass transfer and light irradiation in the scaled-up reactor. We preliminarily propose immobilizing the photocatalyst on the wall of a non-opaque reactor, which can be irradiated from outside and increase the light harvest (Fig. S20a). Meanwhile, the biomass stuff, stored in a cage, is located at the bottom of the reactor and heated to extract the lignin *in situ*. In this way, the efficient light irradiation and isolation of photocatalysts can be readily achieved. Alternatively, the photocatalyst

could be modified with ferromagnetic materials, and after the reaction, the photocatalyst can be separated from the solid residue in the magnetic field (Fig. S20b). However, the problems of mass transfer and light irradiation could remain. In addition, a flow reactor is designed based on the reaction mechanism which may overcome the difficulty in mass transfer and light irradiation (Fig. S20(c)). This flow reactor was composed of three units, namely the aldehyde generation unit, the lignin extraction and protection unit, and the photothermal transformation unit. In this way, the photocatalyst and the biomass are located in different device units which may overcome the difficulty in mass transfer and light utilization. The installation and feasibility tests of these devices are underway in our lab.

Fig. S20 | The proposed reactors and catalyst modification for efficient photothermal catalytic transformation of lignin in biomass. (a) The batch reactor with immobilization of photocatalyst on the wall. (b) Modification of photocatalyst with magnetic materials to isolate photocatalyst under magnetic field. (c) The flow reactor divides the aldehyde generation, lignin extraction, and photothermal reaction into different units.

Reviewer #2:

This article mainly developed a photothermal catalytic process for lignin conversion using Pd/TiO₂ under UV light at 140 °C, which can efficiently convert birch sawdust to phenolic monomers with a 40% yield in 8 hours. The 1,3-diol protection strategy is both very interesting and powerful. I think this manuscript could be accepted after minor modifications. The following suggestions could help to improve this manuscript.

Q1. Although photothermal catalysis has certain advantages over traditional thermal catalysis and photocatalysis, this article lacks comparison with similar work performance and does not highlight its advantages.

Response: Thank you for your suggestion. We reviewed the traditional thermal catalytic and photocatalytic transfer hydrogenation of lignin and made a comparison with our work which is shown in Table S1.

It can be found that the adopted temperature in our work is lower than most of the published thermal-catalytic systems for the transfer hydrogenolysis of protolignin. The novelty of the concept of lignin transformation was also shown in the revised Figure 1. We added the following discussion in the revised introduction part.

‘Compared to the monomer yields from reported works focusing on the lignin and protolignin valorization via transfer hydrogenolysis, this photothermal system delivered an ideal monomer yield that was close to the theoretical value but under a milder condition (Table S1, for more information).’

‘In brief, we realized the efficient protolignin conversion to aromatics via the photothermal catalytic transfer hydrogenolysis intensified by the in-situ extraction-protection strategy to overcome the shortcomings of single thermal catalysis or photocatalysis.’

Table S1 | Comparison of photothermal catalytic transfer hydrogenolysis of protolignin with previous representative works on thermal-catalytic and photocatalytic transfer hydrogenolysis of lignin

Entry	Substrate	Catalyst	T (°C)	h ν	Solvent	t (h)	Monomers (wt%)	Main products	Ref.
1	birch	Ni ₅₀ Pd ₅₀ /SBA-15	245	-	2-PrOH /H ₂ O	4	37	4-propylsyringol, 4-propenylsyringol, 4-propylguaiacol, 4-propenylguaiacol	1
2	poplar	Pd/C	225	-	MeOH	3	28	propyl and ethyl-substituted monomers	2
3	poplar	Pd/C	225	-	ethylene glycol	3	21	propyl and ethyl-substituted monomers	3
4	Swedish birch	Pd/C	210	-	EtOH/H ₂ O	2	40	4-propylsyringol, 4-propenylsyringol	4
5	birch	Ni/C	200	-	MeOH	6	54	4-propylsyringol, 4-propylguaiacol	5
6	Hemp Hurd	Pd/C+p- toluenesulfonic acid	200	-	MeOH/H ₂ O (HCOOH)	4	38	monophenols	6
7	birch	Co-phen/C	200	-	EtOH/H ₂ O (5 equiv. HCOOH and 5 equiv. HCOONa)	4	34	4-propylsyringol, 4-propenylsyringol, 4-propylguaiacol, 4-propenylguaiacol	7
8	birch	Pt/C	190 ^a	-	MeOH/ H ₂ O	3	29	4-propyl syringol, 4-propenyl syringol	8
9	birch	Ru/C	190	-	EG (10 wt% choline chloride)	8	59	propylphenol	9
10	poplar	Ru/C+H ₂ SO ₄	185-195	-	ethylene glycol	6	27	4-propylsyringol, 4-propylguaiacol	10
11	poplar	Pd-PdO/TiO ₂	180	-	H ₂ O (STH)	6	40	monophenols	11
12	Bagasse	Pd/AC+H ₄ SiW ₁₂ O ₄₀	170	-	2-PrOH	5	35	4-ethylphenol, 4-ethylguaiacol	12
13	birch	Pt/NiAl ₂ O ₄	140	-	H ₂ O	24	47	propyl and ethyl-substituted monomers	13
14	poplar	rhodium terpyridine complexes	110	-	H ₂ O (STH)	12	17	Aromatic monomers	14

15	Rice-straw lignin	NiMo-MACS	340	-	formic acid/EtOH	6	72	oil	15
16	technical lignin	-	300	-	EtOH	4	21	Phenolic monomers	16
17	Kraft lignin	Co/carbon nanotube	280	-	EtOH	0.5	66	bio-oil	17
18	organosolv poplar lignin	Ni ₁₀ Cu ₅ /C	270	-	EtOH/2-PrOH	4	63	propyl and ethyl-substituted monomers	18
19	cornstalk hydrolysis residue	Ru/AC	260	-	Ethyl acetate/H ₂ O	5	43	aromatics	19
20	acid-extracted birch lignin	PtRe/TiO ₂	240	-	2-PrOH/ H ₂ O	12	19	monophenols	20
21	organosolv poplar lignin	ReO _x /AC	200	-	2-PrOH	8	11	Phenolic monomers	21
22	organosolv birch lignin	PdNi ₄ /MIL-100 (Fe)	180	-	H ₂ O (STH)	6	17	Guaiacol, 4-methoxyacetophenone	22
23	dioxasolv beech lignin	Ni/Al ₂ O ₃ -600	170	-	2-PrOH	12	13	monomers	23
24	birch	thiol-capped ultrathin ZnIn ₂ S ₄ microbelts	25~40	450 nm	CH ₃ CN/H ₂ O	8	29	Syringyl and guaiacyl -derived ketones	24
25	birch	CdS quantum dots	r.t.	420-780 nm	MeOH/H ₂ O	8	27	Syringyl and guaiacyl -derived ketones	25
26	dioxanesolv poplar lignin	ZnIn ₂ S ₄	42	455 nm	Acetone/2-PrOH	24	10	monophenols	26
27	birch	Pd/TiO₂/HCl	140	370 nm	EtOH/dioxane	8	40	4-propylsyringol, 4-propylguaiacol	This work
28	birch	Pd/TiO₂/HCl	focused sunlight		EtOH/dioxane	6	34	4-propylsyringol, 4-propylguaiacol	This work

Note: HD, hydrogen donor; STH, self-hydrogen transfer. (a) microwave.

(a) Hydrogen-free reductive (thermal) catalytic fractionation for lignin-first biorefining

😊 Fast in-situ lignin extraction; High monomers yield. ☹️ Harsh conditions; Possible lignin recondensation.

(b) Photocatalytic self-transfer hydrogenolysis of lignin in lignocellulose

😊 Mild conditions; No need of external hydrogen donor. ☹️ Difficult lignin extraction; Low monomers yield and selectivity.

(c) This work: Photothermal catalytic transfer hydrogenolysis of protolignin via the *in-situ* protection

😊 Solar-driven photothermal catalysis; Relative mild conditions; Controllable in-situ lignin extraction; High monomers yield.

Fig. 1 | A comparison of the developed thermal catalytic method, photocatalytic method, and our proposed photothermal catalytic method for transfer hydrogenolysis of protolignin in lignocellulose.

Q2. HCl may play multiple roles in the reaction system, but the exploration and description in the article are not scientific and detailed enough.

Response: Thank you for your suggestion. The addition of HCl has a favorable effect on the selective ether bond cleavage, in situ lignin extraction, and in situ aldehyde protection of the 1,3-diol motif in lignin. The impact on the selective ether bond cleavage has been proved by the controlled photothermal experiment of model **1** without HCl (Fig. 3c), in which no *n*-propylbenzene was generated and the benzyl alcohol (**7**) and 2-(2-methoxyphenoxy)ethan-1-ol (**8**) became the main products. Meanwhile, the lignin was barely extracted from birch sawdust without HCl under the heating conditions. Therefore, HCl was vital for the *in situ* lignin extraction. Besides, the condensation of lignin 1,3-diol motif with acetaldehyde can be improved by HCl. However, HCl has a negative effect on the hemicellulose and even cellulose retaining. The photothermal experiment of birch sawdust without the photocatalyst addition but with HCl indicated that hemicellulose was totally decomposed and cellulose was partly decomposed. The multiple roles of HCl are illustrated in the following Figure S15. And we added the following discussion on the various roles of HCl in the main text.

‘In addition, HCl may play multiple roles during the photothermal catalytic conversion of protolignin (Fig. S15). The above tests on the lignin model have proved the promoting effect of HCl on the selective ether bond cleavage (Fig. 3C). As for the lignocellulosic substrate, the lignin was barely extracted from birch sawdust without HCl under the heating conditions. Therefore, HCl was vital for the in-situ lignin extraction. Besides, the condensation of lignin 1,3-diol motif with acetaldehyde can be improved by HCl. However, HCl has a negative effect on the hemicellulose and even cellulose retaining (vide infra). The photothermal experiment of birch sawdust without the addition of photocatalysts but with HCl indicated that hemicellulose was totally decomposed and cellulose was partly decomposed (Table S6).’

Table S6 | The analysis of small products after transfer hydrogenolysis of birch sawdust

Reaction samples		With HCl	With Pd/TiO ₂ and HCl
Lignin products/wt%		0	7
		0	23
	others	0	10
Hemicellulose products/wt%	lactic acid	36	20
	xylose	22	10
	arabinose	27	10
	levulinic acid	1	2
	acetic acid	1	1
Residue cellulose/wt%		66	59

Conditions: birch sawdust (60 mg), 3%-Pd/TiO₂ (0 or 10 mg), EtOH (1 mL), dioxane (0.6 mL), HCl (37%, 10 μL), Kessil LED (370 nm), 140 °C, argon atmosphere. The yields of lignin products were determined by GC. The yields of hemicellulose products were determined by HPLC. The quantification of residue cellulose was determined by the amount of glucose obtained from the further acid-catalyzed hydrolysis of birch solid residue after the photothermal reaction.

Fig. S15 | Multiple roles of HCl during photothermal catalytic transformation of lignocellulose.

Q3. The analysis of biomass products is particularly complex, while the analytical methods here are not sufficiently clear, such as lacking details of the calculation formulas for productivity and yield.

Response: Thank you for your suggestions. The analysis methods were supplemented in the Method of revised text in the manuscript. All aromatic products were quantified using gas chromatography (Shimadzu GC-2014) equipped with an HP-5 capillary column (30 m × 0.32 mm × 0.25 μm) and a flame ionization detector. The injector and detector temperatures were set at 280 °C. The column temperature was initially maintained at 80°C for 2 min, then heated to 260°C at a rate of 10 °C min⁻¹, and then maintained for another 8 min. Molar yields of products from the lignin model reaction were calculated as

$$Yield (i) = \frac{n(i)}{n(s)} \times 100\%$$

where $n(i)$ and $n(s)$ are the mole of product i and the initial lignin model substrate, respectively. Mass yields of aromatic monomer products from biomass reaction were calculated as

$$Yield (i) = \frac{m(i)}{m(b)c(l)} \times 100\%$$

where $m(i)$, $m(b)$, and $c(l)$ are the mass of product i , the mass of initial biomass, and the content of lignin determined by the NREL method, respectively. The water-soluble depolymerized products from hemicellulose were quantified using high-performance liquid chromatography (Shimadzu LC-20AT) equipped with a C18 column. Mass yields of aliphatic monomer products from biomass reaction were calculated as

$$Yield (i) = \frac{m(i)}{m(b)c(h)} \times 100\%$$

where $m(i)$, $m(b)$, and $c(h)$ are the mass of product i , the mass of initial biomass, and the content of hemicellulose determined by the NREL method, respectively.

Reviewer #3:

Li and coauthors reported direct conversion of protolignin to aromatics via the photothermal catalytic transfer hydrogenolysis process intensified by the in-situ protection strategy. The authors found that a 40 wt.% yield of phenolic monomers can be obtained when the depolymerization of birch sawdust was performed with ethanol as the hydrogen donor in 8 h, which can be ascribed to the 1,3-diol-protection between the formed C=O and the hydroxyl groups in the side chain of benzene ring. This work provided a new idea for lignin conversion. However, some issues should be carefully addressed, so I cannot recommend its publication on Nature Commun.

Q1. This work uses diluted HCl in the biomass conversion, however, it should be noticed that diluted HCl itself is a good catalyst for the cleavage of β -O-4 in lignin. Furthermore, HCl is a catalyst for the conversion of cellulose and hemicellulose. Did the author check the performance of lignin/lignin model conversion only with HCl? Besides the monophenols, did the authors find other small molecular products? For example, furfural, HMF, and ethyl levulinate? if not, what are the final states for cellulose and hemicellulose?

Response: Thank you for your questions.

No reaction of lignin model **1** with HCl but without photocatalyst was observed (Figure 3b). Meanwhile, no phenolic monomers were obtained from the reaction of birch with only HCl. Therefore, the acidolysis of β -O-4 to release monomers did not occur in under certain thermal or photothermal conditions. However, the single HCl induced the hydrolysis of hemicellulose in biomass. The main products from hemicellulose were lactic acid, xylose, and arabinose, and a trace amount of levulinic acid and acetic acid was also obtained based on the HPLC analysis (Table S6). Cellulose underwent partial decomposition, and polysaccharides may be the products because no small molecules were observed.

Besides, we thoroughly concluded the multiple roles of HCl during the photothermal catalytic transformation of lignocellulose. The addition of HCl has a

favorable effect on the selective ether bond cleavage, in situ lignin extraction, and in situ aldehyde protection of the 1,3-diol motif in lignin. The effect on the selective ether bond cleavage has been proved by the controlled photothermal experiment of model **1** without HCl (Fig. 3c), in which no *n*-propylbenzene was generated and the benzyl alcohol (**7**) and 2-(2-methoxyphenoxy)ethan-1-ol (**8**) became the main products. Meanwhile, the lignin was barely extracted from birch sawdust without HCl under the heating conditions. Therefore, HCl was vital for the *in-situ* lignin extraction. Besides, the condensation of lignin 1,3-diol motif with acetaldehyde can be improved by HCl. However, HCl has a negative effect on the hemicellulose and even cellulose retaining. The photothermal experiment of birch sawdust without the addition of photocatalyst but with HCl indicated that hemicellulose was totally decomposed and cellulose was partly decomposed. The multiple roles of HCl are illustrated in the following figure. And we added the above discussion on multiple roles of HCl to the main text.

‘In addition, HCl may play multiple roles during the photothermal catalytic conversion of protolignin (Fig. S15). The above tests on the lignin model have proved the promoting effect of HCl on the selective ether bond cleavage (Fig. 3C). As for lignocellulosic substrate, the lignin was barely extracted from birch sawdust without HCl under the heating conditions. Therefore, HCl was vital for the in-situ lignin extraction. Besides, the condensation of lignin 1,3-diol motif with acetaldehyde can be improved by HCl. However, HCl has a negative effect on the hemicellulose and even cellulose retaining (vide infra). The photothermal experiment of birch sawdust without the addition of photocatalysts but with HCl indicated that hemicellulose was totally decomposed and cellulose was partly decomposed (Table S6).’

Table S6 The analysis of small products after transfer hydrogenolysis of birch sawdust

Reaction samples		With HCl	With Pd/TiO ₂ and HCl
Lignin products/wt%		0	7
		0	23
	others	0	10
Hemicellulose products/wt%	lactic acid	36	20
	xylose	22	10
	arabinose	27	10
	levulinic acid	1	2
	acetic acid	1	1
Residue cellulose/wt%		66	59

Conditions: birch sawdust (60 mg), 3%-Pd/TiO₂ (0 or 10 mg), EtOH (1 mL), dioxane (0.6 mL), HCl (37%, 10 μL), Kessil LED (370 nm), 140 °C, argon atmosphere. The yields of lignin products were determined by GC. The yields of hemicellulose products were determined by HPLC. The quantification of residue cellulose was determined by the amount of glucose obtained from the further acid-catalyzed hydrolysis of birch solid residue after the photothermal reaction.

Fig. S15 | Multiple roles of HCl during photothermal catalytic transformation of lignocellulose.

Q2. Fig. 2a, it seems that lots of reactant was formed by-products? What are the side-reactions and by-products? In general, metallic Pt has a comparable catalytic activity as Pd in many CTH reactions, however, the acetal 2 was the main products in the Pt/TiO₂ catalytic system, while the monophenols are the primary in the presence of Pd/TiO₂. Please explain this.

Response: Thank you for your questions. We divided this comment into two questions.

About the by-product. We repeated the TiO₂-catalyzed transformation of model 1 under photothermal conditions, only a 20% conversion of model 1 and a 4% yield of product 2 were achieved. We revised this data in fig. 2a-entry 1. The GC data of TiO₂ and 3%-Pd/TiO₂ catalyzed the transformation of model 1 under photothermal conditions were provided in the following figures. Besides of the products 2-5, no other by-products were observed. The following discussion was added to the manuscript. *‘No other byproducts can be identified in the gas chromatography data (Fig. S3-S4).’*

About the activity of Pt-based catalyst. Normally the metallic Pt has a stronger catalytic hydrogenation/hydrogenolysis activity than Pd in the CTH reactions. Our previous works on photocatalytic CTH (Green Chem. 2023, 25, 6869–6880; Green Chem., 2020, 22, 3802-3808; ChemCatChem, 2022, 14, e202200120) demonstrated that Pt/TiO₂ was the efficient catalyst for hydrogenation of aryl rings and hydrogenolysis of aryl C–O bonds, while Pd/TiO₂ was effective in the hydrogenolysis of benzylic C–O bonds and C=O bonds. However, we also observed that the metallic Pt was sensitive to aldehyde byproducts from the oxidation of primary alcohols which were used as the hydrogen donor. For example, the hydrogenolysis performance of diaryl ether in 2-PrOH was much better than those in methanol, ethanol, and 1-PrOH (Green Chem. 2023, 25, 6869–6880, figure 1e). The addition of a certain amount of formaldehyde or acetaldehyde in 2-PrOH totally inhibited the transfer hydrogenolysis activity of Pt/TiO₂. In the current work, the acetaldehyde was generated from ethanol oxidation which may restrain the its hydrogenolysis ability.

Fig. S3 | The GC data of TiO_2 catalyzed the transformation of model 1 under photothermal conditions

Fig. S4 | The GC data of 3%-Pd/ TiO_2 catalyzed the transformation of model 1 under photothermal conditions

Q3. Also Fig. 2a, the increase of 3%-Pd/ TiO_2 dosage resulted in an increase of guaiacol yield, but, with 6%-Pd/ TiO_2 , both the yields of products 3 and 4 were remarkably decreased. This seems unreasonable. The author claimed that this is because of the critical synergistic effect between the semiconductor TiO_2 and loading Pd NPs. This explanation is too superficial, can the author state this clearly?

Response: Thank you for your suggestion.

To reveal the reason for the poor activity of 6%-Pd/ TiO_2 compared to 3%-Pd/ TiO_2 , detailed characterizations on morphology and photo-responsive properties were performed. The SEM images indicated that the loading of different amounts of Pd did not affect the aggregation state of TiO_2 particles. The BET surface area slightly

decreased after loading of Pd particles (Table S2). The TEM images indicated that the different loading amounts dramatically affected the size of Pd particles. The mean size of Pd particles in 3%-Pd/TiO₂ was 5.1 nm, while that in 6%-Pd/TiO₂ was 8.1 nm. The larger size of metal particles will induce a low percentage of surface atoms. Therefore, the higher loading amount of Pd may not induce more active metal sites. The UV-vis absorption suggested that the loading of Pd particles induced the visible light absorption of the photocatalyst. The photocurrent test indicated that the loading of 3% metallic Pd slightly increased the photocurrent, however, the loading of 6% metallic Pd dramatically decreased the photocurrent. Normally, the loaded metal on the semiconductor is beneficial to charge-carrier separation because metal sites could accommodate the photo-generated electron due to their low Fermi level. Herein we deduced that the 6% loading of Pd may affect the light absorption of TiO₂ support which could induce the observed low photocurrent compared to bare TiO₂. Therefore, a higher Pd loading amount may not induce more exposed active sites due to the increase of Pd particle mean size to 8.1 nm (Fig. S6). Meanwhile, the larger size of Pd particles could affect the light absorption of semiconductor support and decrease the density of photo-generated electrons (Fig. S8). The following discussion was added to the manuscript and the following table and figures were added to the supporting information.

‘The higher Pd loading amount may not induce more exposed active sites due to the increase of Pd particle mean size to 8.1 nm (Fig. S6). Meanwhile, the larger size of Pd particles could affect the light absorption of semiconductor support and decrease the density of photo-generated electrons (Fig. S8).’

Fig. S6 | The SEM (a) and TEM images (b, c) of 3%-Pd/TiO₂, and SEM (d) and TEM images (e, f) of 6%-Pd/TiO₂.

Table S2 | The BET analysis of photocatalysts

Sample	BET surface area (m ² /g)
TiO ₂	63.9
3%-Pd/TiO ₂	47.7
6%-Pd/TiO ₂	46.4

Fig. S8 | The photo-responsive properties of photocatalysts. (a) UV-vis absorption. (b) fluorescence spectra ($\lambda_{EX} = 300$ nm). (c) Photocurrent test (under 370 nm LED irradiation).

Q4. Line 217, “especially for the HCl, which could inhibit the oxidative cleavage of the C–C bond” . This should be carefully stated. In general, a Brønsted acid favors the cleavage of C–C in the oxidative depolymerization of lignin.

Response: Thank you for your suggestion. The cleavage of the C–C bond without acid additives may be originated from the β -scission after the generation of benzylic oxygen radical which was not an oxidative process (Chem. Rev. 2023, 123, 8, 4510–4601). We delete this sentence in the revised manuscript.

Q5. Fig. 6, the yield of monophenols in i-PrOH is significantly lower than that in PrOH. The author attributed this to the relatively lower activity of acetone than propaldehyde. Surely, this would be a reason for this. However, the dehydrogenation capability of i-PrOH is substantially greater than PrOH, furthermore, the active H* species are crucial for the cleavage of the C–O in lignin. To be more conceivable, I suggest the author investigate this using model compounds.

Response: Thank you for your suggestion.

Thank you for your suggestion. The dehydrogenation capability of 2-PrOH is truly greater than 1-PrOH. The model compound test in different alcohols indicated the secondary alcohols (2-PrOH and 2-BuOH) delivered higher yields of hydrogenolysis products than primary alcohols (1-PrOH and 1-BuOH). These results proved your proposal, namely the importance of active H* species for the cleavage of C–O bonds. As for protolignin transformation, however, the hydrogenolysis of birch sawdust in primary alcohols delivered much higher monomer yields than those in secondary alcohols (30-42% versus 16-18 wt%). This phenomenon was contrary to the hydrogenolysis result of the lignin model 1 in different alcohols. Meanwhile, the externally extracted lignin with 1,3-diol protection delivered a higher yield of monomers than externally extracted lignin without 1,3-diol protection (11 wt% versus 4 wt%). Therefore, the higher yields of monomers from protolignin conversion in primary alcohol with modest hydrogen donating ability demonstrated the importance

of protecting effect in this tandem extraction/hydrogenation process of real biomass. Back to the hydrogenolysis of lignin models, the hydrogenolysis performance of the ethylidene acetal-protected model was even slightly worse than that of the unprotected model (Figure 2c). Thus, the ability of hydrogen donating rather than protecting effect dominated the model conversion. These results also indicated the difference between lignin model transformation with protolignin transformation.

The following discussion was added to the manuscript. ‘*This phenomenon was contrary to the hydrogenolysis result of the lignin model 1 in different alcohols (Fig. S9). Even the photocatalytic dehydrogenation product ketone from secondary alcohol was a modest protection reagent, the secondary alcohol was a better hydrogen donor than primary alcohol. Besides, the hydrogenolysis performance of the ethylidene acetal-protected model was slightly worse than that of the unprotected model (Fig. 2b and Fig. 2c). Thus, the ability of hydrogen donating rather than protecting effect dominated the model conversion. In contrast, the higher yields of monomers from protolignin conversion in primary alcohol with modest hydrogen donating ability demonstrated the importance of protecting effect in this tandem extraction/hydrogenation process of real biomass⁴⁹. These results also indicated the difference between lignin model transformation with protolignin transformation..*

Fig. S9 | The photothermal catalytic transfer hydrogenolysis of lignin model in different alcohols

Q6. Many logical and grammatical errors are shown, please recheck the manuscript.

Response: Thank you for your suggestion. We thoroughly checked the manuscript and fixed these logical and grammatical errors.

REVIEWERS' COMMENTS

Reviewer #1

I am pleased to see that the authors have well addressed the issues raised by the reviewers. One interesting phenomenon is that the photocurrent density of the optimal photocatalyst increased with temperature (Figure S12), which means that increasing the temperature was beneficial for the separation of photogenerated carriers. This is rare for TiO₂ photocatalysis. Can the authors explain this further? Also, I would like to see future work by the authors on Ni with this developed photothermal catalytic process. I would recommend its publication in Nature Communications.

Response: Thank you for your kind words and thoughtful comments on our work.

Regarding heating-promoted charge carriers separation, there are two possible reasons. The first reason is that the heating provides the kinetic energy for the separation and transportation of charge carriers. Ho's work also observed the heating-induced increase of photocurrent (Figure 4f in *Angew. Chem. Int. Ed.* 2019, 58, 3077-3081). Another reason is that the heating accelerates the matter transportation in solution, thus the consumption of photo-generated carriers may be promoted which induces the promotion of charge carriers separation (*Applied Catalysis B: Environmental* 2019, 243, 760-770).

In addition, thank you again for your suggestion on Ni-based photothermal catalytic lignin conversion. The related research is underway in our lab.

Reviewer #2 (Remarks to the Author):

I think the authors have fully answered my questions, and the work supported the conclusions and claims. I accept this work as it is.

Response: We thank the reviewer for the kind comments.

Reviewer #3:

Surely, this work had been promoted after the revision. However, many key problems have not been solved, thus, I cannot recommend its publication on Nature Commun. This work used a conventional photocatalyst of Pd/TiO₂ for the photothermal catalytic transfer hydrogenolysis of protolignin. The authors found that, 40 wt% yield of phenolic monomer was obtained at 140 °C for 8 h when ethanol was used as the hydrogen donor. This monophenol yield was three times higher than the extracted 1,3-diol-protected lignin's conversion. This result more likely relates to the resource of lignin, for example, the formation of recalcitrant C–C bond in lignin separation, rather than the advantage of this cascade technique of photothermal conversion. Furthermore, the author claimed that the C_β–OAr bond before other C–O bonds, leading to a high yield of phenolic monomer products in this photothermal process. This is also just a consensus in lignin conversion. The same conclusion can also be observed by the acetalation of the side-chain of lignin with the primary alcohols. Moreover, the potential hydrogen supply of the OCH₃ in lignin should also be discussed. Thus, both the technical advance and the scientific finding are insufficient to support its publication on this high-level journal.

Response: Thank the reviewer for the comments.

(1) About the higher monomer yields from protolignin than extracted protected lignin.

The extraction of protected lignin was performed according to the reported method in Luterbacher's works (Science 2016, 354, 329-333; Angew. Chem. Int. Ed. 2018, 57, 1356-1360). The 2D HQSC NMR spectra in our manuscript clearly demonstrated the formation of ethylidene acetal-protected β-O-4 linkage. Therefore, the formation of a recalcitrant C–C bond barely occurred under this condition. We deduced that the difficulty in transformation of extracted lignin was ascribed to the large size of lignin fragments which was verified by the GPC analysis in the manuscript.

(2) For the claiming on the prior cleavage of C_β–OAr bond before other C–O bonds which provide a high yield of phenolic monomer products in this photothermal

process, this phenomenon is general in the lignin depolymerization. However, the photothermal-catalytic-effect-induced prior cleavage of C β -OAr bond was reported for the first time. The developed photothermal catalytic method, especially the solar-driven photothermal conversion in this work, may inspire more strategies to achieve value-added utilization of lignin.

- (3) About the acetalation of the side chain of lignin with the primary alcohols. The lignin extraction with 1,3-diol protection and following hydrogenolysis were performed in two pots in previous works. In this manuscript, the photocatalytic oxidation of primary alcohol to aldehyde, the acetalation of side-chain during in situ extraction, and the hydrogenolysis were performed in one pot. We think the reported acetalation protection strategy for lignin conversion was updated in our work.
- (4) About the potential hydrogen supply of the OCH $_3$ in lignin. No de-methoxylated products were observed in the transformation of the lignin model and real lignin. Therefore, the hydrogen supply of OCH $_3$ can be excluded from our work.